# Lactate biosensors for spectrally and spatially multiplexed fluorescence imaging

Yusuke Nasu [1,2] ✉, Abhi Aggarwal [3,4], Giang N. T. Le [1,5], Camilla Trang Vo[6], Yuki Kambe [7], Xinxing Wang[8], Felix R. M. Beinlich [6], Ashley Bomin Lee[6], Tina R. Ram[9], Fangying Wang[9], Kelsea A. Gorzo[9], Yuki Kamijo[1], Marc Boisvert[10,11], Suguru Nishinami[12], Genki Kawamura[1], Takeaki Ozawa [1], Hirofumi Toda[12], Grant R. Gordon [9], Shaoyu Ge [8], Hajime Hirase [6,13], Maiken Nedergaard [6,13], Marie-Eve Paquet [10,11], Mikhail Drobizhev [14], Kaspar Podgorski [3,4] & Robert E. Campbell [1,10,11] ✉

L-Lactate is increasingly appreciated as a key metabolite and signaling molecule in mammals. However, investigations of the inter- and intra-cellular dynamics of L-lactate are currently hampered by the limited selection and performance of L-lactate-specific genetically encoded biosensors. Here we now report a spectrally and functionally orthogonal pair of high-performance genetically encoded biosensors: a green fluorescent extracellular L-lactate biosensor, designated eLACCO2.1, and a red fluorescent intracellular L-lactate biosensor, designated R-iLACCO1. eLACCO2.1 exhibits excellent membrane localization and robust fluorescence response. To the best of our knowledge, R-iLACCO1 and its affinity variants exhibit larger fluorescence responses than any previously reported intracellular L-lactate biosensor. We demonstrate spectrally and spatially multiplexed imaging of L-lactate dynamics by coexpression of eLACCO2.1 and R-iLACCO1 in cultured cells, and in vivo imaging of extracellular and intracellular L-lactate dynamics in mice.

L-Lactate was once considered a waste by-product of glucose metabolism[1]. However, growing evidence suggests that L-lactate plays a variety of important roles as both an energy source and a signaling molecule in the nervous system[2], tumor microenvironment[3], gut microbiome[4], and immune system[5]. These roles have impacts on physiological and pathological processes over scales and environments ranging from subcellular[6], to intercellular[2,7], to interorgan[8].

Investigations of the emerging roles of L-lactate in cells and tissues would be facilitated by a set of genetically encoded fluorescent biosensors that could enable high resolution spatially and

[1]Department of Chemistry, School of Science, The University of Tokyo, Bunkyo-ku, Tokyo 113-0033, Japan. [2]PRESTO, Japan Science and Technology Agency, Chiyoda-ku, Tokyo 102-0075, Japan. [3]Janelia Research Campus, Howard Hughes Medical Institute, Ashburn, VA 20147, USA. [4]Allen Institute for Neural Dynamics, Seattle, WA 98109, USA. [5]Department of Chemistry, University of Toronto, Toronto, ON M5S 3H6, Canada. [6]Center for Translational Neuromedicine, Faculty of Health and Medical Sciences, University of Copenhagen, Copenhagen 2200, Denmark. [7]Department of Pharmacology, Graduate School of Medical and Dental Science, Kagoshima University, Sakuragaoka, Kagoshima 890-8544, Japan. [8]Department of Neurobiology and Behavior, Stony Brook University, Stony Brook, NY 11794, USA. [9]Hotchkiss Brain Institute, Cumming School of Medicine, Department of Physiology and Pharmacology, University of Calgary, Calgary, AB T2N 4N1, Canada. [10]CERVO Brain Research Centre, Québec, QC G1J 2G3, Canada. [11]Department of Biochemistry, Microbiology and Bioinformatics, Laval University, Québec, QC G1E 1T2, Canada. [12]International Institute for Integrative Sleep Medicine, University of Tsukuba, Tsukuba, Ibaraki 305-8575, Japan. [13]Center for Translational Neuromedicine, University of Rochester Medical Center, Rochester, NY 14642, USA. [14]Department of Microbiology and Cell Biology, Montana State Universities, Bozeman, MT 59717, USA. ✉e-mail: nasu@chem.s.u-tokyo.ac.jp; campbell@chem.s.u-tokyo.ac.jp

temporally resolved imaging of L-lactate in both extracellular and intracellular spaces. Towards this end, we previously developed a genetically encoded green fluorescent biosensor, designated eLACCO1.1, for extracellular L-lactate[9]. Others have reported cyan or green fluorescent genetically encoded biosensors for imaging intracellular L-lactate using fluorescence intensity[10-15] or fluorescence lifetime[16]. However, the first-in-class eLACCO1.1 suffers from aggregation and limited fluorescence response on the surface of mammalian cells. Another limitation of the current set of L-lactate biosensors is that no red fluorescent biosensors for intracellular

L-lactate have yet been reported, hindering imaging with other green fluorescent biosensors in spatially and spectrally multiplexed manner.

We now report a spectrally and functionally orthogonal pair of high-performance genetically encoded biosensors: the second-generation green fluorescent eLACCO2.1 for extracellular L-lactate and the first-generation red fluorescent R-iLACCO1 for intracellular L-lactate (Fig. 1a, b). This biosensor pair enables robust imaging of extracellular and intracellular L-lactate in cultured mammalian cells and living mice.

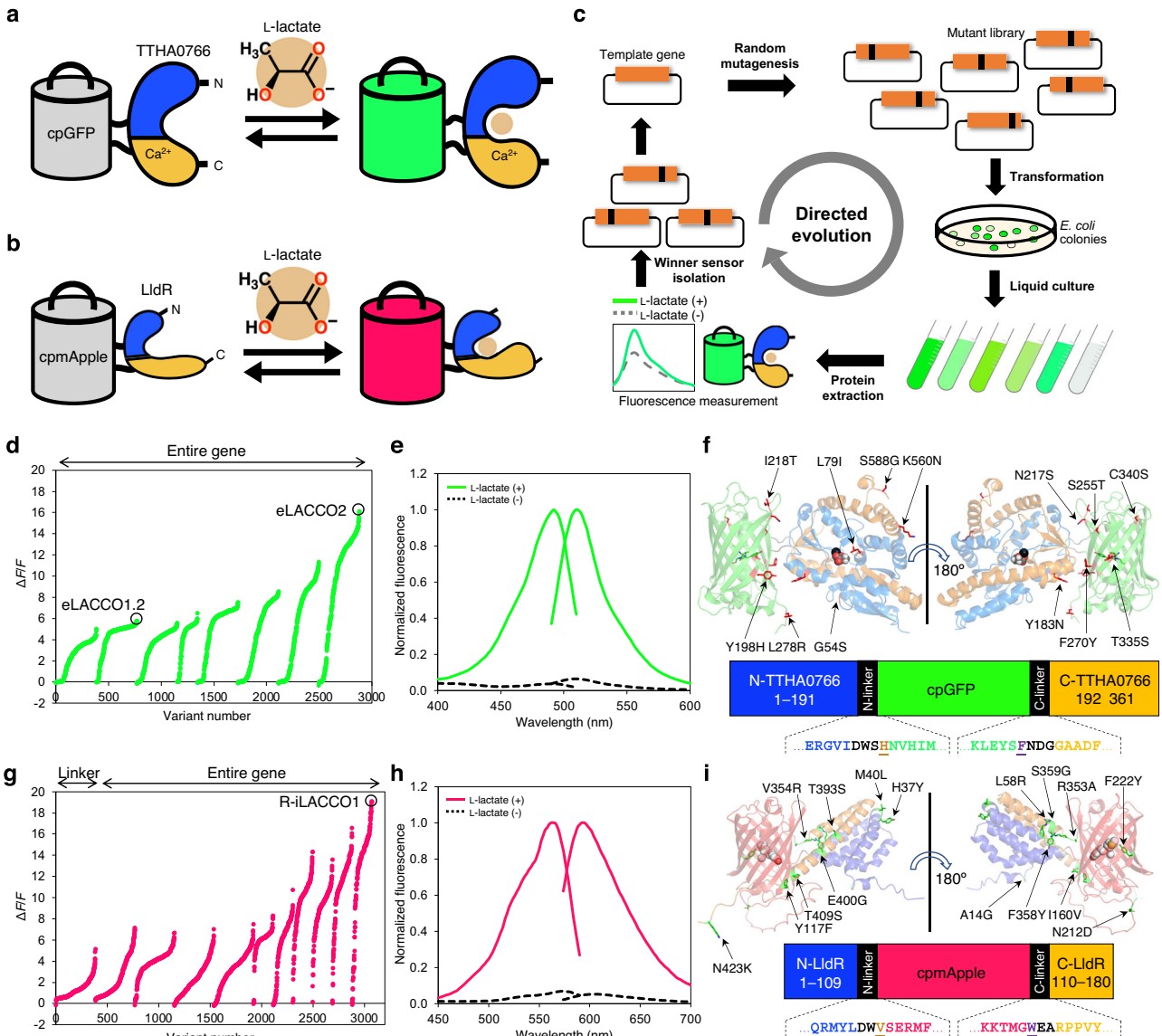

**Fig. 1 | Development of eLACCO2.1 and R-iLACCO1. a** Schematic representation of TTHA0766-based eLACCO and its mechanism of response to L-lactate. **b** Schematic representation of LldR-based R-iLACCO and its mechanism of response to L-lactate. **c** Schematic of directed evolution workflow. Specific sites (i.e., the linkers) or the entire gene of template L-lactate biosensor were randomly mutated and the resulting mutant library was used to transform *E. coli*. Bright colonies were picked and cultured, and then proteins were extracted to examine $\Delta F/F$ upon addition of 10 mM L-lactate. A mixture of the variants with the highest $\Delta F/F$ was used as the template for the next round. **d** $\Delta F/F$ rank plot representing all crude proteins tested during the directed evolution of eLACCO. For each round, tested variants are ranked from lowest to highest $\Delta F/F$ value from left to right. **e** Excitation and emission spectra of purified eLACCO2.1 in the presence (10 mM) and absence of L-

lactate. **f** Overall representation of the eLACCO1 crystal structure (PDB ID: 7E9Y [https://www.rcsb.org/structure/7E9Y]) with the position of mutations indicated. L-Lactate and Ca²⁺ (black) are shown in a sphere representation. In the primary structure of eLACCO2.1 (bottom), linker regions are shown in black and the two "gate post" residues[31] in cpGFP are highlighted in dark orange (His195) and purple (Phe437). **g** $\Delta F/F$ rank plot representing all crude proteins tested during the directed evolution of R-iLACCO. **h** Excitation and emission spectra of purified R-iLACCO1 in the presence (10 mM) and absence of L-lactate. **i** Overall representation of the R-iLACCO1 model structure[41] with the position of mutations indicated. In the primary structure of R-iLACCO1 (bottom), linker regions are shown in black and the two "gate post" residues[31] in cpmApple are highlighted in dark orange (Val112) and purple (Trp351). Source data are provided as a Source Data file.

**Table 1 | One- and two-photon photophysical parameters of eLACCO2.1**

| L-Lactate | eLACCO2.1 | | eLACCO1.1 (ref. 9) | | iLACCO1 (ref. 19) | | jGCaMP8s (ref. 48) | | EGFP |
|---|---|---|---|---|---|---|---|---|---|
| | — | + | — | + | — | + | —[a] | +[a] | |
| Relative fraction of neutral chromophore ($\rho_N$) | 0.97 | 0.24 | 0.97 | 0.57 | 0.97 | 0.79 | 0.74[b] | 0.17[b] | 0.07[c] |
| Relative fraction of anionic chromophore ($\rho_A$) | 0.03 | 0.76 | N/A | 0.43 | N/A | 0.21 | 0.26[b] | 0.83[b] | 0.93[c] |
| Neutral extinction coefficient (mM$^{-1}$ cm$^{-1}$, $\varepsilon_N$) | 42 | 45 | 34 | 38 | 39 | 39 | N/A | N/A | N/A |
| Anionic extinction coefficient (mM$^{-1}$ cm$^{-1}$, $\varepsilon_A$) | 76 | 65 | 78 | 89 | 81 | 74 | 2.1 | 57 | 56.0[d]; 58.3[e] |
| Neutral quantum yield ($\varphi_N$) | 0.06 | N/A | 0.20 | 0.23 | N/A | N/A | N/A | N/A | N/A |
| Anionic quantum yield ($\varphi_A$) | 0.47 | 0.82 | 0.60 | 0.78 | 0.72 | 0.84 | 0.47 | 0.53 | 0.67[d]; 0.76[e] |
| Molecular brightness ($\rho_A \times \varepsilon_A \times \varphi_A$) | 1.1 | 41 | 4.5[f] | 30 | 0.74 | 13 | 0.26 | 25 | 34.9[d]; 41.2[e] |
| Neutral one-photon absorbance peak (nm) | 395 | 398 | 398 | 397 | 400 | 400 | N/A | N/A | N/A |
| Anionic one-photon absorbance peak (nm) | 493 | 494 | 493 | 496 | 492 | 492 | N/A | N/A | 488[d] |
| Two-photon cross section (GM) (position/nm) | N/A | 32 (932) | N/A | 48 (924) | N/A | 42 (928) | N/A | 25 | 54 (911)[e] |
| Two-photon brightness $F_2$ (GM) (position/nm)[g] | 0.88[h] (932) | 20 (932) | 2.2 (924) | 16 (924) | 0.52 (928) | 7.3 (928) | N/A | N/A | 41 (911)[e] |

*N/A* not applicable.

[a]Ca$^{2+}$.

[b]Calculated from an equation, $\log_{10}$([anionic]/[neutral]) = pH - p$K_a$, where pH is 7.2 and p$K_a$ is 7.65 (Ca$^{2+}$ free) and 6.51 (Ca$^{2+}$ bound).

[c]Calculated from an equation, $\log_{10}$([anionic]/[neutral]) = pH - p$K_a$, where pH is 7.2 and p$K_a$ is 6.1 (ref. 49).

[d]Data from ref. 49.

[e]Data from ref. 50.

[f]Estimated from the independently measured $F_{1,bound}/F_{1,free}$ ratio at 500 nm.

[g]Two-photon brightness is calculated as $F_2 = \rho_A \times s_{2,A} \times \varphi_A$ at the peak wavelength (presented in parentheses), where $s_{2,A}$ is the peak two-photon absorption cross section.

[h]Estimated from the independently measured $F_{2,bound}/F_{2,free}$ ratio at 932 nm.

## Results

### Development of a second-generation green fluorescent L-lactate biosensor, eLACCO2.1

To engineer a second-generation green fluorescent extracellular L-lactate biosensor, we first identified eLACCO0.9 as a promising candidate due to a faster fluorescence response than eLACCO1 (Supplementary Fig. 1). To develop variants of eLACCO0.9 with larger change in fluorescence intensity ($\Delta F/F = (F_{max} - F_{min}) / F_{min}$) upon L-lactate treatment, we performed directed evolution with screening of crude protein extracts for $\Delta F/F$ (Fig. 1c). The first two rounds of evolution used libraries created by random mutagenesis of the entire gene of eLACCO0.9 ($\Delta F/F = 3.9$) and resulted in the identification of the eLACCO1.2 with $\Delta F/F$ of 5.8 (Fig. 1d). To avoid potential complications due to disulfide bond formation[17], we introduced the Cys340Ser mutation to produce eLACCO1.3 ($\Delta F/F = 4.8$, crude protein extract). Six additional rounds of directed evolution led to the eLACCO2 variant with $\Delta F/F$ of 16 (Fig. 1d).

Characterization of purified eLACCO2 revealed an affinity for L-lactate (apparent $K_d$ ~ 280 μM) much higher than typical extracellular concentrations, which are in the millimolar range (Supplementary Fig. 2). To tune the affinity for extracellular L-lactate, we mutated residues in the L-lactate binding pocket (Supplementary Fig. 2b). Of ten variants investigated, eLACCO2 Leu79Ile, designated eLACCO2.1, exhibited a lower affinity (apparent $K_d = 1.9$ mM) with a minimal decrease in $\Delta F/F$ compared to eLACCO2 (Supplementary Fig. 2b, c). eLACCO2.1 has a $\Delta F/F$ of 14 as a purified protein upon treatment with 10 mM L-lactate, which is 3.5-fold higher than eLACCO1.1 (Fig. 1e and Supplementary Table 1), and has 13 mutations relative to eLACCO0.9 (Fig. 1f and Supplementary Fig. 3). A non-responsive control biosensor, designated deLACCO1, was engineered by incorporating the Asp444Asn mutation into eLACCO2 to abolish L-lactate binding, followed by three rounds of directed evolution to improve fluorescence brightness (Supplementary Figs. 3–5). We thoroughly characterized eLACCO2.1 in a soluble form (Table 1, Supplementary Fig. 6, Supplementary Table 1, and Supplementary Note 1).

### Development of a red fluorescent biosensor for intracellular L-lactate, R-iLACCO1

To construct a red fluorescent L-lactate biosensor (Fig. 1b), we inserted a circularly permuted red fluorescent protein (cpmApple), derived from the TTHA0766-based red fluorescent L-lactate biosensor R-eLACCO1 (ref. 18), into the L-lactate binding domain (LBD) of the *Escherichia coli* LldR at position 109 (Supplementary Fig. 7a)[19]. Relative to Ca$^{2+}$-dependent TTHA0766, LldR has no Ca$^{2+}$ dependence and is functional in the cytosol where the Ca$^{2+}$ concentration is low (~0.1–1 μM)[10,19]. The resulting variant, designated R-iLACCO0.1, exhibited dim fluorescence and only a slight increase in fluorescence intensity ($\Delta F/F = 0.7$, crude protein extract) upon treatment with L-lactate (Supplementary Fig. 7b). To improve R-iLACCO0.1, we optimized the lengths of linkers that connect cpmApple and LldR, and found that a variant (designated R-iLACCO0.2) with two amino acids of the C-terminal linker deleted had substantially improved brightness, while retaining a comparable fluorescence response ($\Delta F/F = 0.6$, crude protein extract) (Supplementary Fig. 7c, d). To develop variants of R-iLACCO0.2 with larger $\Delta F/F$, we performed one round of C-terminal linker optimization, followed by ten rounds of directed evolution by random mutagenesis of the whole gene (Fig. 1c, g). This effort produced R-iLACCO1 with $\Delta F/F$ of 20 (purified protein, Fig. 1h). R-iLACCO1 contains a total of 16 mutations relative to R-iLACCO0.1 (Fig. 1i and Supplementary Fig. 8).

The wide range of physiological concentrations of intracellular L-lactate inspired us to engineer lower affinity variants of R-iLACCO1. In the final round of directed evolution, we screened all randomly picked variants in the absence of L-lactate, at 0.5 mM L-lactate, and at 10 mM L-lactate, to assess their apparent $K_d$ values. This effort led us to identify two lower affinity variants, designated R-iLACCO1.1 and R-iLACCO1.2 (Supplementary Fig. 9). To engineer a non-responsive control biosensor, designated R-diLACCO1, we incorporated the Asp69Asn mutation into R-iLACCO1 and then performed two rounds of directed evolution to improve fluorescence brightness (Supplementary Figs. 4, 5, and 8). We undertook a detailed characterization of the biochemical and spectral properties of R-iLACCO1 and its affinity variants (Supplementary Figs. 10 and 11, Table 2, Supplementary Table 2, and Supplementary Note 2).

### Optimization of cell surface localization

Our previous work on eLACCO1.1 and R-eLACCO2 had demonstrated that the exact combination of N-terminal leader sequence and C-terminal anchor domain plays a critical role in determining the

**Table 2 | One- and two-photon photophysical parameters of R-iLACCO1**

| L-Lactate | R-iLACCO1 | | R-eLACCO2 (ref. 18) | | R-GECO1 (ref. 35) | | mApple (ref. 49) |
|---|---|---|---|---|---|---|---|
| | − | + | − | + | −[a] | +[a] | |
| Relative fraction of neutral chromophore ($\rho_N$) | 0.95 | 0.39 | 0.88 | 0.35 | 0.94[b] | 0.18[b] | 0.17[c] |
| Relative fraction of anionic chromophore ($\rho_A$) | N/A | 0.61 | 0.12 | 0.65 | 0.06[b] | 0.82[b] | 0.83[c] |
| Neutral extinction coefficient (mM$^{-1}$ cm$^{-1}$, $\varepsilon_N$) | 36 | 55 | 29 | N/A | 33 | 30 | N/A |
| Anionic extinction coefficient (mM$^{-1}$ cm$^{-1}$, $\varepsilon_A$) | 84 | 88 | 13 | 59 | 89 | 69 | 75 |
| Neutral quantum yield ($\varphi_N$) | N/A | N/A | N/A | N/A | N/A | N/A | N/A |
| Anionic quantum yield ($\varphi_A$) | 0.17 | 0.21 | 0.11 | 0.28 | 0.15 | 0.21 | 0.46 |
| Molecular brightness ($\rho_A \times \varepsilon_A \times \varphi_A$) | N/A | 11.3 | 0.17 | 10.7 | 0.80 | 12 | 28.6 |
| Neutral one-photon absorbance peak (nm) | 444 | 447 | 444 | ND | N/A | N/A | N/A |
| Anionic one-photon absorbance peak (nm) | 579 | 561 | 578 | 567 | 577 | 563 | N/A |
| Two-photon cross section (GM) (position/nm) | 28 (1080) | 72 (1048) | 35 (1080) 236 (740) | 51 (1060) 125 (740) | 24 (1072) | 31 (1056) | N/A |
| Two-photon brightness $F_2$ (GM) (position/nm) | N/A | 9.2 (1048) | 0.46 (1080) 3.1 (740) | 9.3 (1060) 23 (740) | 0.21 (1072) | 5 (1056) | N/A |

*N/A* not applicable, *ND* not determined.

[a]Ca$^{2+}$.

[b]Calculated from an equation, $\log_{10}$([anionic]/[neutral]) = pH - p$K_a$, where pH is 7.2 and p$K_a$ is 7.69 (Ca$^{2+}$ free) and 6.36 (Ca$^{2+}$ bound).

[c]Calculated from an equation, $\log_{10}$([anionic]/[neutral]) = pH - p$K_a$, where pH is 7.2 and p$K_a$ is 6.5.

efficiency of cell surface targeting, as well as the functionality, of a particular extracellular biosensor[9,18]. We assessed 23 leaders[18] and 10 anchors[20] with eLACCO1.1 to identify the optimal combination for use in primary neurons (Fig. 2a). We first examined the efficiency of cell surface targeting with human CD59-derived leader sequence and 10 different anchors, including peptide-based and lipid (glycosylphosphatidylinositol, GPI)-based anchors (Fig. 2b). Fluorescence imaging revealed that the reticulon-4 receptor (Nogo receptor, NGR) GPI anchor resulted in the highest efficiency of cell surface localization and also exhibited a higher Δ*F*/*F* compared to the original CD59-eLACCO1.1-CD59 (Fig. 2b, c)[9]. We next screened various N-terminal leader sequences in combination with the NGR anchor in the presence of L-lactate (Fig. 2d). Of 23 leader sequences examined, influenza hemagglutinin (HA) provided very good membrane localization and also showed greater fluorescent signal relative to CD59-eLACCO1.1-NGR in neurites (Fig. 2e). Having identified HA and NGR as the optimal combination for cell surface targeting, we appended these sequences to eLACCO2.1 to produce our final construct. HA-eLACCO2.1-NGR was used for all cell imaging experiments (Fig. 2a).

### Characterization of eLACCO2.1 in live mammalian cells and acute brain slices

We expressed and characterized HA-eLACCO2.1-NGR in mammalian cells. eLACCO2.1 localized well on the cell membrane (Fig. 3a), while CD59-eLACCO1.1-CD59 formed some fluorescent puncta as previously reported[9]. Application of 10 mM L-lactate robustly increased fluorescence intensity (Δ*F*/*F* of 8.8 ± 0.2, mean ± s.e.m.) of eLACCO2.1 expressed on HeLa cells (Fig. 3b, c). This response is 138% that of eLACCO1.1 (Δ*F*/*F* of 6.4 ± 0.2, mean ± s.e.m.) (Fig. 3c). The control biosensor deLACCO1 had good membrane localization and, as expected, did not respond to L-lactate (Fig. 3a–d). eLACCO2.1 has an in situ apparent $K_d$ of 580 μM for L-lactate (Fig. 3d). Similar to eLACCO1.1 (ref. 9), eLACCO2.1 displays Ca$^{2+}$ dependent fluorescence with an apparent $K_d$ of 270 μM, which is much lower than the extracellular Ca$^{2+}$ concentration in brain tissue (1.5–1.7 mM)[21] and serum (0.9–1.3 mM)[22] (Fig. 3e). To assess the response kinetics of eLACCO2.1, we exposed eLACCO2.1-expressing HeLa cells to a solution containing 0 mM, then 10 mM, and then 0 mM L-lactate. eLACCO2.1 had faster on and off rates ($\tau_{on}$ of 57 ± 4 s and $\tau_{off}$ of 20 ± 2 s, mean ± s.d.) than eLACCO1.1 ($\tau_{on}$ of 74 ± 7 s and $\tau_{off}$ of 24 ± 5 s, mean ± s.d.) (Fig. 3f). The photostability of eLACCO2.1 ($\tau_{bleach}$ of 189 ± 26 s, mean ± s.e.m.) was comparable to the parent EGFP ($\tau_{bleach}$ of 169 ± 4 s, mean ± s.e.m.) and higher than eLACCO1.1 ($\tau_{bleach}$ of 111 ± 6 s, mean ± s.e.m.) in the absence of L-lactate, and was lower ($\tau_{bleach}$ of 55 ± 1 s, mean ± s.e.m.) than eLACCO1.1 ($\tau_{bleach}$ of 83 ± 7 s, mean ± s.e.m.)

in the presence of L-lactate (Fig. 3g). deLACCO1 was less photostable than EGFP in both the presence ($\tau_{bleach}$ of 74 ± 1 s, mean ± s.e.m.) and absence ($\tau_{bleach}$ of 70 ± 1 s, mean ± s.e.m.) of L-lactate.

To characterize the performance of eLACCO2.1 on the surface of neurons, we expressed it under the control of human synapsin (hSyn) promoter in rat primary cortical and hippocampal neurons. HA-eLACCO2.1-NGR exhibited bright membrane-localized fluorescence in neurons, while CD59-eLACCO1.1-CD59 exhibited membrane-localized fluorescence with some puncta (Fig. 3h). Upon bath application of 10 mM L-lactate, eLACCO2.1 had Δ*F*/*F* values of 8.1 ± 0.7 (mean ± s.e.m.), which is 5.4-fold higher than eLACCO1.1 (Δ*F*/*F* = 1.5 ± 0.2, mean ± s.e.m.) (Fig. 3i, j). To determine whether eLACCO2.1 could be used to detect L-lactate concentration changes on the surface of astrocytes, we expressed it under the control of the gfaABC1D promoter in rat primary cortical and hippocampal astrocytes. The biosensor was expressed well (Fig. 3h), and displayed improved Δ*F*/*F* values of 7.3 ± 1.0 (mean ± s.e.m., Fig. 3i, k) compared to eLACCO1.1 (Δ*F*/*F* = 4.9 ± 0.5, mean ± s.e.m.), upon bath application of 10 mM L-lactate. deLACCO1 showed no response to L-lactate, similar to our results obtained with HeLa cells and neurons. Taken together, these results indicated that eLACCO2.1 with the optimized leader and anchor has improved performance relative to eLACCO1.1 and can be used for imaging of extracellular L-lactate concentration dynamics on the surfaces of neurons and astrocytes.

To determine whether eLACCO2.1 could be used to detect changes in endogenous L-lactate concentration in intact tissue, we transfected the mouse neocortex with adeno-associated virus (AAV) encoding HA-eLACCO2.1-NGR under the control of hSyn promoter to express it on the surface of neurons (Fig. 3l). In the ex vivo context of brain slices, an aglycemic challenge (that is, depletion of glucose) caused the fluorescence of eLACCO2.1 to decrease (Δ*F*/*F* = −0.20 ± 0.04, mean ± s.e.m., Fig. 3m). This response is consistent with the uptake of available extracellular L-lactate[23]. High-frequency electrical afferent stimulation has been reported to elevate the concentration of extracellular L-lactate around neurons[24]. Indeed, theta burst stimulation caused a significant elevation in the eLACCO2.1 signal that did not subside after 15 min of recording (Δ*F*/*F* = 0.11 ± 0.02, mean ± s.e.m., Fig. 3n). Collectively, these results indicated that eLACCO2.1 can detect endogenous changes in extracellular L-lactate in intact tissue.

### Characterization of R-iLACCO1 in live mammalian cells

Next, we characterized R-iLACCO variants expressed in mammalian cells in comparison with previously reported intracellular L-lactate

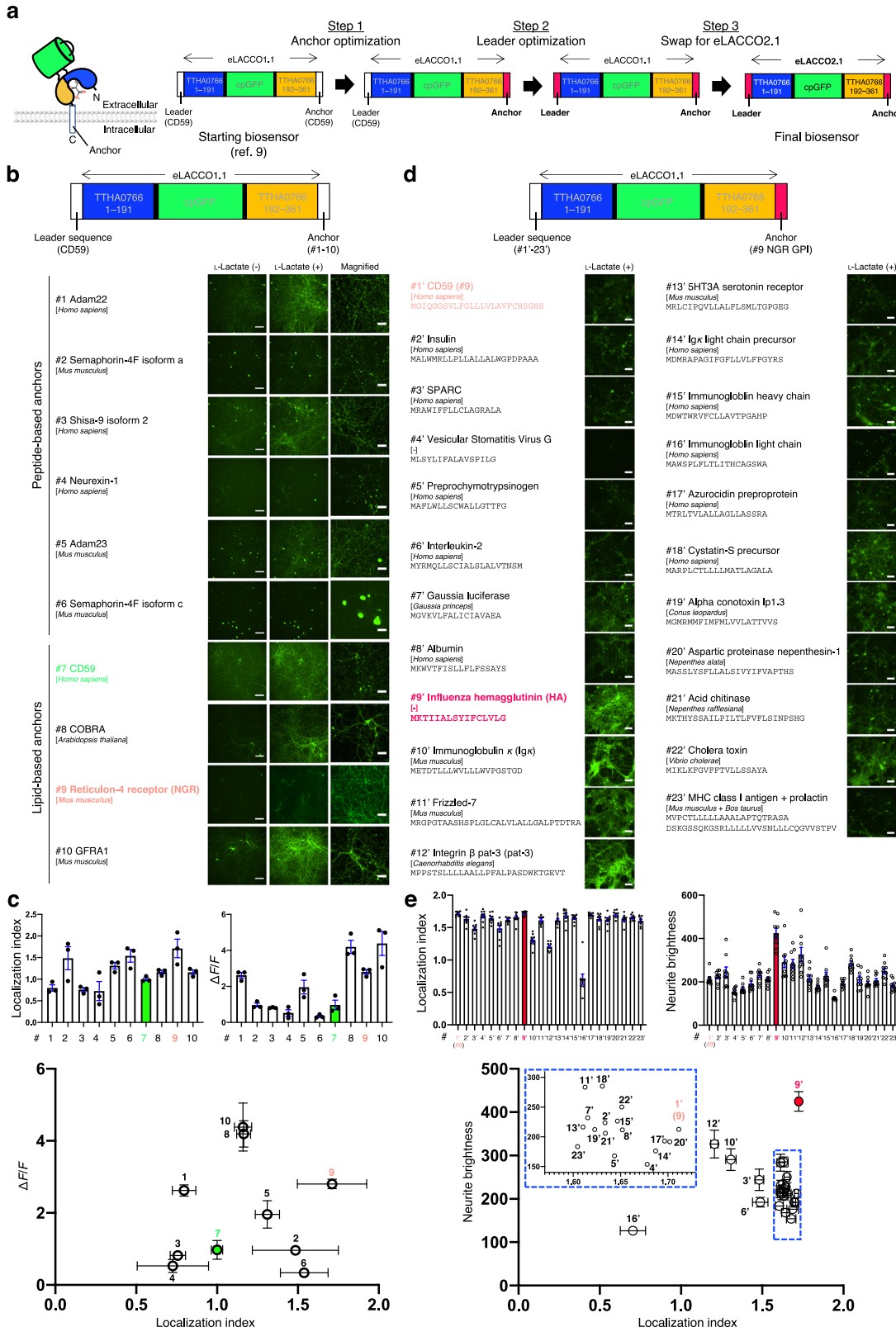

**Fig. 2 | Optimization of leader/anchor combination for cell-surface expression.**
**a** Overview of biosensor optimization. **b** Localization of eLACCO1.1 with CD59
leader sequence and different anchors in rat primary neurons. Scale bars, 200 μm.
Right panels show magnified images in the presence of L-lactate. Scale bars, 50 μm.
**c** Localization index and Δ*F/F* of eLACCO1.1 with CD59 leader sequence and dif-
ferent anchors in rat primary neurons. Mean ± s.e.m., *n* = 3 field of views (FOVs) over

three wells per construct. **d** Localization of eLACCO1.1 with different leader
sequences and NGR GPI anchor in rat primary neurons in the presence of L-lactate.
Scale bars, 200 μm. **e** Localization index and neurite brightness of eLACCO1.1 with
different leader sequences and NGR GPI anchor in rat primary neurons. Mean ±
s.e.m., *n* = 9 FOVs over three wells per construct. Source data are provided as a
Source Data file.

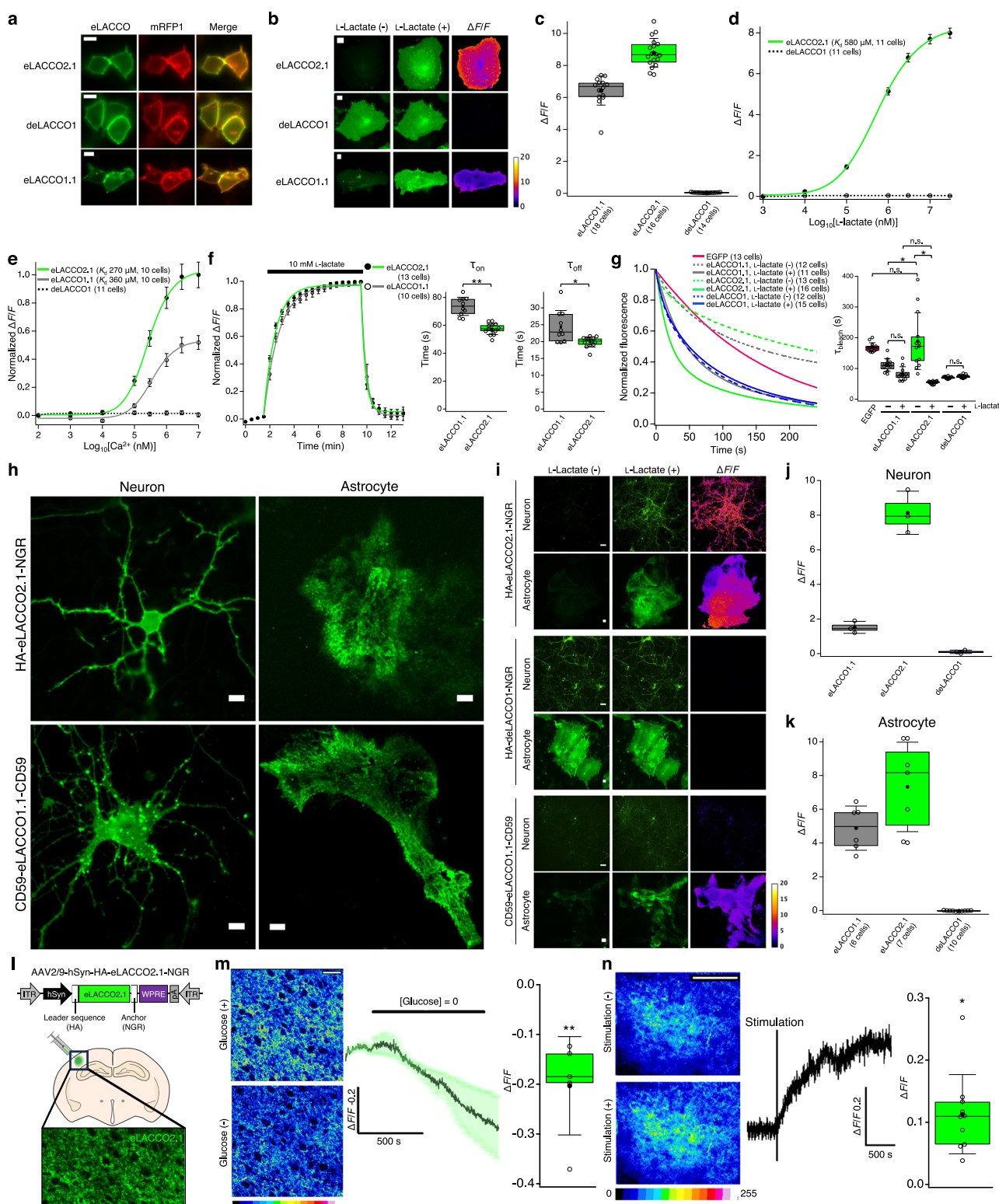

biosensors. The three R-iLACCO affinity variants all robustly increased fluorescence intensity ($\Delta F/F$ of 5.5 ± 0.6, 7.6 ± 0.7, and 11 ± 0.5, for R-iLACCO1, R-iLACCO1.1, and R-iLACCO1.2, respectively, mean ± s.e.m.) upon treatment of HeLa cells with L-lactate (Fig. 4a, b). Under the same conditions, the control biosensor R-diLACCO1 did not respond to L-lactate. R-iLACCO1, R-iLACCO1.1, and R-iLACCO1.2 have in situ apparent $K_d$s of 680 μM, 3.0 mM, and 4.0 mM for L-lactate, respectively (Fig. 4c). Bath application of L-lactate revealed that R-iLACCO1.1 ($\tau_{on}$ of 23.1 ± 0.7 s, $\tau_{off}$ of 66.5 ± 7.5 s, mean ± s.e.m.) and R-iLACCO1.2 ($\tau_{on}$ of

23.2 ± 0.3 s, $\tau_{off}$ of 45.4 ± 5.1 s, mean ± s.e.m.) showed a slower on rate and a faster off rate than R-iLACCO1 ($\tau_{on}$ of 15.8 ± 0.2 s, $\tau_{off}$ of 174 ± 21 s, mean ± s.e.m.) (Fig. 4d).

Many cpmApple-based biosensors undergo photoactivation when illuminated with blue light[25], hampering their combined use with blue light-responsive tools such as GCaMP[26] and channelrhodopsin (ChR)[27]. We found that blue-light illumination of R-iLACCO variants (~10 mW cm⁻² at 470 nm) in HeLa cells, elicited a small increase ($\Delta F/F$ ~ 0.09) in fluorescence intensity (Fig. 4e). This result indicates

**Fig. 3 | Characterization of eLACCO2.1 in live mammalian cells and acute brain slices. a** Localization of eLACCO2.1 and control biosensor deLACCO1 with the optimized leader sequence and anchor, and eLACCO1.1 with CD59-derived leader and anchor domain[9] in HEK293T cells. mRFP1 (ref. 51) with the Igκ leader sequence and PDGFR transmembrane domain was used as a cell surface marker. Scale bars, 10 μm. **b** Expression on cell surface of HeLa cells before and after 10 mM ʟ-lactate stimulation. Scale bars, 10 μm. **c** Δ$F/F$ on HeLa cells. **d** In situ ʟ-lactate titration. Mean ± s.e.m. **e** In situ $Ca^{2+}$ titration in the presence (10 mM) of ʟ-lactate. Mean ± s.e.m. **f** Time course of the fluorescence response. Mean ± s.d. Two-tailed Student's *t* test. *$P = 0.0043$, **$P < 0.0001$. **g** Photobleaching curves (left) and $\tau_{bleach}$ (right). Mean ± s.e.m. One-way ANOVA followed by Tukey's multiple comparison. *$P < 0.0001$. **h** Expression of eLACCO variants expressed on the surface of rat primary neurons and astrocytes. Scale bars, 10 μm. **i** ʟ-Lactate (10 mM) response of eLACCO variants expressed on cell surface of rat primary neurons and astrocytes.

Scale bars, 10 μm. **j** Δ$F/F$ of eLACCO biosensors under hSyn promoter on rat primary neurons. $n = 3$ FOVs over three wells per construct. **k** Δ$F/F$ of eLACCO biosensors under gfaABC1D promoter expressed on rat cortical astrocytes. **l** Schematic illustration of AAV injection into the somatosensory cortex. Image shows expression of hSyn-HA-eLACCO2.1-NGR. **m** Two-photon imaging of extracellular ʟ-lactate around neurons expressing in hSyn-HA-eLACCO2.1-NGR, in response to a zero-glucose challenge in acute brain slices. Pseudo coloured images, summary trace (mean ± s.e.m.), and summary data are shown. $n = 5$ slices, **$P = 0.0093$, two-tailed unpaired Student's *t* test compared with vehicle control experiment. Scale bar, 50 μm. **n** Two-photon imaging of extracellular ʟ-lactate around neurons in response to electrical afferent stimulation (theta burst) in acute brain slices. Pseudo coloured images, a representative trace, and summary data are shown. $n = 10$ slices, *$P = 0.0191$, two-tailed paired Student's *t* test comparing peak to baseline. Scale bar, 100 μm. Source data are provided as a Source Data file.

that the photoactivation and recovery of the R-iLACCO variants occurs with much faster kinetics and with far smaller Δ$F/F$ than the maximum ʟ-lactate dependent fluorescence response (Fig. 4b, d). To assess the photostability of R-iLACCO variants, we used the integrated fluorescence (IF) in HeLa cells (rather than to $\tau_{bleach}$), to better account for the complex photobleaching decays of mApple-based biosensors[28]. R-iLACCO1.2 and R-diLACCO1 exhibited photostability (IF of $666 \pm 50$ and $650 \pm 20$, mean ± s.d., respectively) comparable to the parent mApple RFP (IF of $653 \pm 17$, mean ± s.d.), whereas R-iLACCO1 and R-iLACCO1.1 exhibited lower photostability (IF of $530 \pm 20$ and $520 \pm 68$, mean ± s.d., respectively) than mApple (Fig. 4f).

To investigate whether R-iLACCO variants can enable monitoring of the dynamics of endogenous ʟ-lactate in mammalian cells, we performed live cell imaging of R-iLACCO variants in a variety of cell types and stimulation conditions, and did a side-by-side comparison with previously reported ʟ-lactate biosensors (Fig. 4g–j, Table 3, and Supplementary Fig. 12). Glucose treatment promotes glycolysis in glucose-starved cells, leading to an increase of intracellular ʟ-lactate concentration[29] and a corresponding increase in fluorescence intensity for R-iLACCO variants (Fig. 4g). Notably, the low-affinity R-iLACCO1.2 showed the largest fluorescence response of all R-iLACCO variants and its Δ$F/F$ ($2.5 \pm 0.2$, mean ± s.e.m.) is much larger than Δ$F/F$ or Δ$R/R$ of other ʟ-lactate biosensors. Inhibition of monocarboxylate transporter (MCT) suppresses efflux of intracellular ʟ-lactate, resulting in an increase in intracellular ʟ-lactate concentration[30]. The treatment of HeLa cells with MCT inhibitor AR-C155858 evoked robust fluorescence changes of R-iLACCO variants (Δ$F/F$ of $0.08 \pm 0.01$, $0.19 \pm 0.03$, and $0.30 \pm 0.02$, for R-iLACCO1, R-iLACCO1.1, and R-iLACCO1.2, respectively, mean ± s.e.m.) (Fig. 4h). The low-affinity variant R-iLACCO1.2 showed higher Δ$F/F$ than all other ʟ-lactate biosensors except iLACCO1. To confirm the functionality of R-iLACCO variants in neural cells, we performed fluorescence imaging of primary neurons and astrocytes. Treatment of neurons and astrocytes with AR-C155858 yielded a robust increase in the fluorescence intensity of R-iLACCO variants (Fig. 4i, j). An increase in intracellular ʟ-lactate concentration is accompanied with a pH decrease (Supplementary Fig. 13), which can cause a fluorescence intensity decrease for FP-based biosensors[31]. However, R-iLACCO variants showed a robust increase in their fluorescence intensity in response to an increase in intracellular ʟ-lactate level in all imaging conditions investigated (Fig. 4 and Supplementary Fig. 14). Overall, these results indicated that R-iLACCO variants could be particularly useful for imaging of intracellular ʟ-lactate concentration dynamics in mammalian cells.

### In vivo imaging of eLACCO2.1 and R-iLACCO1.1 in mice
To validate the performance of eLACCO2.1 in vivo, we measured the fluorescence response of eLACCO2.1 expressed in visual cortex neurons of awake mice upon intracerebroventricular (i.c.v.) injection of ʟ-lactate (Fig. 5a). eLACCO2.1 showed a significant fluorescence

response to the ʟ-lactate in comparison to saline (Fig. 5b), whereas the control biosensor deLACCO1 showed virtually no change in fluorescence intensity (Fig. 5c). These results indicate that eLACCO2.1 is functional as an extracellular ʟ-lactate biosensor in vivo. Next, we aimed to examine the capability of eLACCO2.1 for detecting endogenous ʟ-lactate dynamics in freely moving mice. Systemic injection of insulin leads to a decrease of extracellular ʟ-lactate concentration in mouse brain[32]. We expressed eLACCO2.1 in hippocampal neurons and recorded its fluorescence response upon intraperitoneal (i.p.) injection of insulin (Fig. 5d). eLACCO2.1 showed a significant fluorescence decrease upon insulin treatment in comparison to saline treatment. Moreover, deLACCO1 showed negligible change in fluorescence intensity (Fig. 5e). Thus, eLACCO2.1 can detect the dynamics of endogenous ʟ-lactate in freely moving mice.

Stimulation of mouse whiskers has been reported to evoke an increase of intracellular ʟ-lactate concentration in the somatosensory cortex neurons[33]. To investigate whether the R-iLACCO biosensor enables monitoring of intracellular ʟ-lactate dynamics in vivo, we expressed R-iLACCO1.1 in somatosensory cortex neurons and observed the fluorescence response upon whisker stimulation (Fig. 5f, g). The fluorescence imaging of shallowly anesthetized mice revealed that R-iLACCO1.1 showed a stimulation-dependent fluorescence response eliciting a gradual signal increase (Fig. 5h, i). These results suggest that R-iLACCO1.1 can be used to monitor endogenous ʟ-lactate dynamics in living mice.

### Spectrally and spatially multiplexed ʟ-lactate imaging in glioblastoma cells
To directly observe the production of ʟ-lactate in the cytosol and its export to the extracellular space, we coexpressed eLACCO2.1 and R-iLACCO1.2 for extracellular and intracellular ʟ-lactate imaging in starved glioblastoma cells (Fig. 6a). Glucose stimulation evoked concomitant fluorescence increases of both eLACCO2.1 and R-iLACCO1.2 (Fig. 6b, c), indicating that ʟ-lactate produced in the cytosol is rapidly exported to extracellular space. Notably, the intracellular ʟ-lactate signals showed a transient peak at which the slope of the extracellular ʟ-lactate signal became substantially smaller. Considering that we rarely observed this transient peak in the presence of an MCT inhibitor (Supplementary Fig. 13d, e), this peak is presumably due to the result from the balance between cytosolic production and efflux via MCT.

To investigate whether the ʟ-lactate is shuttled between the cytosol and organelles, we next monitored the ʟ-lactate dynamics in the mitochondrial matrix and endoplasmic reticulum (ER). We targeted R-iLACCO1.2 in the mitochondrial matrix or ER and coexpressed cytosolic iLACCO1 (ref. 19) in starved glioblastoma cells (Fig. 6d–i). The fluorescence intensities of R-iLACCO1.2 in both mitochondrial matrix and ER increased concomitant with those of iLACCO1 upon glucose treatment. These results indicate that ʟ-lactate produced in cytosol is also shuttled to the mitochondrial matrix and ER in addition to extracellular space.

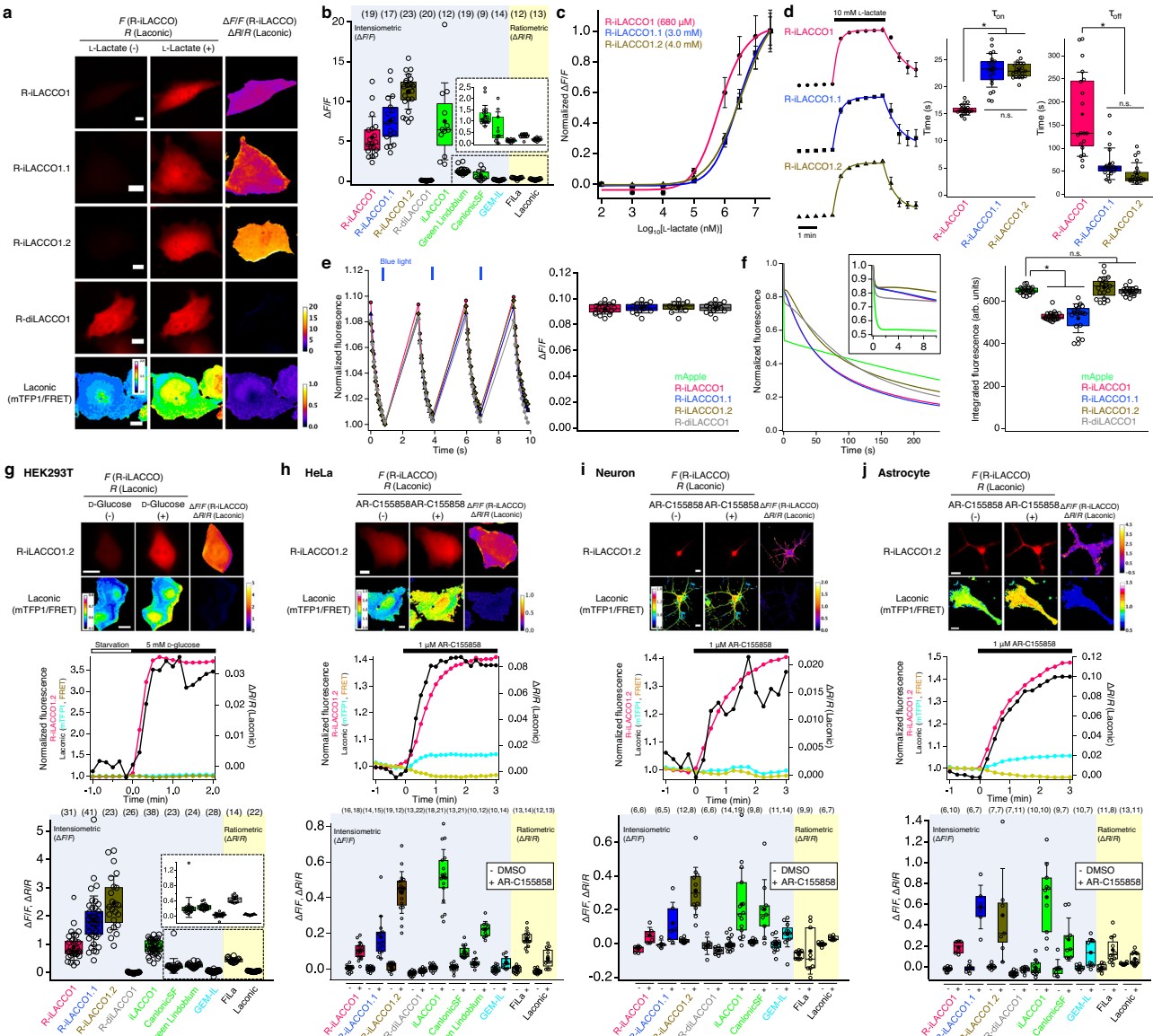

**Fig. 4 | Characterization of R-iLACCO variants in live mammalian cells.**
**a** Fluorescence images in HeLa cells. Scale bars, 10 μm. **b** ΔF/F in HeLa cells in (**a**). Parentheses represent number of cells investigated. **c** In situ L-lactate titration. n = 10, 18, and 18 cells for R-iLACCO1, R-iLACCO1.1, and R-iLACCO1.2, respectively (mean ± s.e.m.). **d** Time course of the fluorescence response in HeLa cells. n = 19, 21, and 21 cells for R-iLACCO1, R-iLACCO1.1, and R-iLACCO1.2, respectively (mean ± s.d.). One-way ANOVA followed by Tukey's multiple comparison. *P < 0.0001.
**e** Fluorescence traces (left) and ΔF/F (right) in response to blue-light illumination of HeLa cells. n = 17, 14, 11, and 14 cells for R-iLACCO1, R-iLACCO1.1, R-iLACCO1.2, and R-diLACCO1, respectively. Fluorescence traces represent mean ± s.e.m.
**f** Photobleaching curves (left, mean ± s.e.m.) and integrated fluorescence (right) in HeLa cells. Inset shows the photobleaching curves in the first 10 s. n = 18, 20, 18, 19, and 17 cells for mApple, R-iLACCO1, R-iLACCO1.1, R-iLACCO1.2, and R-diLACCO1, respectively. One-way ANOVA followed by Tukey's multiple comparison. *P < 0.0001. **g** Representative images (top) and fluorescence traces (middle) in

HEK293T upon 5 mM D-glucose treatment after starvation. Box plots (bottom) show ΔF/F or ΔR/R for R-iLACCO variants and other biosensors. Parentheses represent number of cells investigated. Scale bars, 10 μm. **h** Representative images (top) and fluorescence traces (middle) in HeLa cells upon 1 μM AR-C155858 treatment. Box plots (bottom) show ΔF/F or ΔR/R for R-iLACCO variants and other biosensors. Parentheses represent number of cells investigated. Scale bars, 10 μm.
**i** Representative images (top) and fluorescence traces (middle) in primary neurons upon 1 μM AR-C155858 treatment. Box plots (bottom) show ΔF/F or ΔR/R for R-iLACCO variants and other biosensors. Parentheses represent number of neurons investigated. Scale bars, 20 μm. **j** Representative images (top) and fluorescence traces (middle) in primary astrocytes upon 1 μM AR-C155858 treatment. Box plots (bottom) show ΔF/F or ΔR/R for R-iLACCO variants and other biosensors. Parentheses represent number of astrocytes investigated. Scale bars, 20 μm. Source data are provided as a Source Data file.

## Discussion

Characterization of our previously reported eLACCO1.1 revealed there could be substantial room to improve its ΔF/F by increasing brightness in the L-lactate-bound state (Table 1). Specifically, eLACCO1.1 exhibited only 43% anionic (bright state) chromophore in the L-lactate-bound state. Indeed, directed evolution led to the identification of eLACCO2.1 which has 76% of the anionic chromophore in the L-lactate-bound state and is brighter than eLACCO1.1 under both one- and two-photon

excitation (Table 1). In addition to the biosensor performance as a purified soluble protein, the combination of leader and anchor is critically important for the biosensor functionality in living cells (Fig. 2 and Supplementary Note 3). Optimization of both in vitro performance by directed evolution and the leader/anchor combination by neuron-based screening finally produced the highly optimized biosensor HA-eLACCO2.1-NGR. This second-generation biosensor substantially outperforms CD59-eLACCO1.1-CD59 in terms of membrane localization

**Table 3 | Comparison of performance for intracellular L-lactate biosensors**

| Biosensors | | Intensiometric | | | | | | | | Ratiometric | | Lifetime |
|---|---|---|---|---|---|---|---|---|---|---|---|---|
| | | R-iLACCO1 | R-iLACCO1.1 | R-iLACCO1.2 | R-diLACCO1 | iLACCO1 | CanlonicSF | Green Lindoblum | GEM-IL | FiLa | Laconic | LiLac |
| Fluorescent protein | | RFP (cpmApple) | | | | GFP (cpGFP) | GFP (cpGFP) | GFP | CFP (sfCFP) | YFP (cpYFP) | CFP/YFP (mTFP1/ Venus) | CFP (mTurquoise2) |
| Lactate binding protein | | LldR (DBD deleted) | | | | LldR (DBD deleted) | TTHA0766 | LldR (DBD deleted) | LldR (DBD deleted) | LldR (DBD deleted) | LldR (full length) | TlpC |
| $K_d$ (µM) as a purified protein | | 74 | 230 | 350 | N/A | 360[a] | 300[b] | 30[c] | 661[d] | 130[e] | 8 and 830[f] | 620[g] |
| $\Delta F/F$ (intensiometric) $\Delta R/R$ (ratiometric) $\Delta LT$ (lifetime) | Purified protein | 20 | 22 | 15 | −0.3 | **30**[a] | 1.9[b] | 4.2[c] | 0.8[d] | 15[e] | 0.2[f] | −0.8[g] |
| | HeLa[h] | 5.49 ± 0.60 | 7.63 ± 0.73 | **11.2 ± 0.46** | 0.06 ± 0.01 | 7.50 ± 1.40 | 1.25 ± 0.10 | 0.71 ± 0.23 | 0.13 ± 0.02 | 0.35 ± 0.03 | 0.19 ± 0.02 | NT |
| | HeLa[i] | 0.10 ± 0.01 | 0.18 ± 0.03 | 0.43 ± 0.03 | −0.01 ± 0.00 | **0.52 ± 0.04** | 0.10 ± 0.01 | 0.22 ± 0.01 | 0.04 ± 0.01 | 0.17 ± 0.01 | 0.06 ± 0.02 | NT |
| | HEK293T[j] | 0.91 ± 0.09 | 1.84 ± 0.14 | **2.46 ± 0.20** | 0.01 ± 0.00 | 0.90 ± 0.04 | 0.23 ± 0.05 | 0.24 ± 0.02 | 0.04 ± 0.01 | 0.44 ± 0.02 | 0.04 ± 0.00 | NT |
| | Neuron[j] | 0.05 ± 0.02 | 0.12 ± 0.05 | **0.31 ± 0.04** | −0.04 ± 0.01 | 0.23 ± 0.06 | 0.20 ± 0.06 | ND | 0.05 ± 0.02 | −0.05 ± 0.05 | 0.03 ± 0.01 | NT |
| | Astrocyte[j] | 0.19 ± 0.02 | 0.57 ± 0.09 | 0.49 ± 0.17 | −0.03 ± 0.01 | **0.67 ± 0.10** | 0.26 ± 0.07 | ND | 0.14 ± 0.04 | 0.16 ± 0.04 | 0.07 ± 0.02 | NT |
| Localization pattern | | **Uniform** | **Uniform** | **Uniform** | **Uniform** | **Uniform** | Puncta | Puncta | **Uniform** | **Uniform** | Puncta | NT |
| Other information observed | | | | | | | Fluorescence oscillation | Dim fluorescence | | | | |

*DBD* DNA binding domain, *ND* not detected, *NT* not tested.
Mean ± s.e.m. Bold represents the best performance of the lactate biosensors investigated.
[a]Data from ref. 19.
[b]Data from ref. 11.
[c]Data from ref. 14.
[d]Data from ref. 13.
[e]Data from ref. 12.
[f]Data from ref. 10.
[g]Data from ref. 16.
[h]$\Delta F/F$ measured in HeLa cells pretreated with 500 µM iodoacetate, 10 µM nigericine, and 2 µM rotenone upon addition of 10 mM L-lactate.
[i]$\Delta F/F$ measured upon 1 µM AR-C155858 treatment.
[j]$\Delta F/F$ measured in pre-starved HEK293T cells upon addition of 5 mM D-glucose.

(Fig. 3a, h), $\Delta F/F$ (8.8 vs. 6.4 in HeLa cells, 8.1 vs. 1.5 in cultured neurons, and 7.3 vs. 4.9 in cultured astrocytes, Fig. 3c, j, k), and kinetics ($\tau_{on}$ of 57 s vs. 74 s, $\tau_{off}$ of 20 s vs. 24 s, Fig. 3f). Furthermore, HA-eLACCO2.1-NGR can monitor the dynamics of endogenous L-lactate in freely moving mice (Fig. 5d, e). An important consideration for the application of eLACCO2.1 is its $Ca^{2+}$ dependent fluorescence response. For extracellular applications this should not be a major concern because the apparent $K_d$ of eLACCO2.1 for $Ca^{2+}$ is substantially lower than the physiological or pathological $Ca^{2+}$ concentration range (-1.5–1.7 mM) in the extracellular space of brain tissue (Fig. 3e)[21]. However, the $Ca^{2+}$ concentration in the extracellular environment can transiently decrease to ~1 mM during neural activity evoked by extreme stimulations such as long trains of experimentally induced action potentials[34]. We determined that a change in extracellular $Ca^{2+}$ concentration from 2 mM to 1 mM could cause a −9% and −5% fluorescence intensity for lactate-bound and lactate-unbound eLACCO2.1, respectively (Supplementary Fig. 15).

Since the first report of a genetically encoded red fluorescent $Ca^{2+}$ biosensor R-GECO1, a variety of other red biosensors have been developed[31,35]. However, despite the remarkable progress in the development of such biosensors, the applications of genetically encoded red fluorescent biosensors have been relatively limited compared to green fluorescent biosensors. This limitation can be attributed to the properties such as smaller $\Delta F/F$, dim fluorescence due to low molecular brightness and expression level, photoactivation upon blue-light illumination[25], and subcellular mislocalization (e.g., accumulation in lysosomes)[36]. To overcome these difficulties, we performed extensive directed evolution to obtain variants with higher $\Delta F/F$ and brightness (Fig. 1c). Ultimately, we developed the R-iLACCO1 variant with high molecular brightness (11.3 in L-lactate-bound state, Table 2) and $\Delta F/F$ of 20 that is one of the highest fluorescence responses for red fluorescent biosensor protein, as measured using purified protein (Supplementary Table 2)[18,31]. We also confirmed that R-iLACCO1 and its affinity variants display no undesirable punctate intracellular accumulation (Fig. 4g–j). In addition, the availability of multiple affinity variants may be helpful for investigating intracellular L-lactate dynamics in different cell types with various basal lactate levels[37]. The apparent $K_d$s of R-iLACCO variants in cells (Fig. 4c) are

much higher than those measured with purified proteins (Supplementary Fig. 10b). Considering a change in temperature (37 °C used for cells vs. 25 °C for purified proteins) does not have a substantial impact on apparent $K_d$s of R-iLACCO variants (Supplementary Fig. 16), this discrepancy must be due to molecular crowding or other yet-unidentified environmental differences between the purified proteins in buffer and the protein expressed in mammalian cells. Side-by-side performance comparisons indicate that R-iLACCO1 variants are the best-in-class intracellular L-lactate biosensors for many cell-based applications (Fig. 4 and Table 3). We took advantage of this optimized performance to monitor the dynamics of endogenous L-lactate in vivo (Fig. 5f–i). The fluorescence response ($\Delta F/F \sim 1\%$) of R-iLACCO1.1 in the somatosensory cortex of whisker stimulated mice is comparable to that of Laconic ($\Delta R/R \sim 1\%$) under similar conditions[38]. Based on the fact that the $\Delta F/F$ of R-iLACCO1.1 in cultured cells is much larger than $\Delta R/R$ of Laconic, a substantially larger response could have been expected. The reason for this discrepancy is unclear, but one possible explanation is that the lactate affinity of R-iLACCO1.1 might not be optimal for detecting the changes that occur under these particular stimulation conditions.

Inspired by the emerging roles of L-lactate, we[19] and other groups[10–16] have previously developed cyan or green FP-based biosensors for intracellular L-lactate. To the best of our knowledge, R-iLACCO1 is the first RFP-based intracellular L-lactate biosensor, allowing multiplexed imaging with blue-light activated optogenetic actuators or other green fluorescent biosensors in different subcellular compartments. Multiplexed imaging of R-iLACCO biosensors with green fluorescent biosensors, or combined use with optogenetic tools, should be performed with caution since blue-light photoactivation ($\Delta F/F \sim 9\%$, Fig. 4e) of R-iLACCO biosensors is non-negligible in some applications of cultured cells (Fig. 4h–j) and in vivo (Fig. 5f–i). Parallel experiments with the control biosensor R-diLACCO1 would be recommended in the multiplexed applications. Indeed, we demonstrate that R-iLACCO1.2 in combination with GFP-based eLACCO2.1 or iLACCO1 enables monitoring the L-lactate dynamics in various cell compartments. Remarkably, we found that L-lactate levels in both mitochondria and ER show a slight decrease when the cytosolic

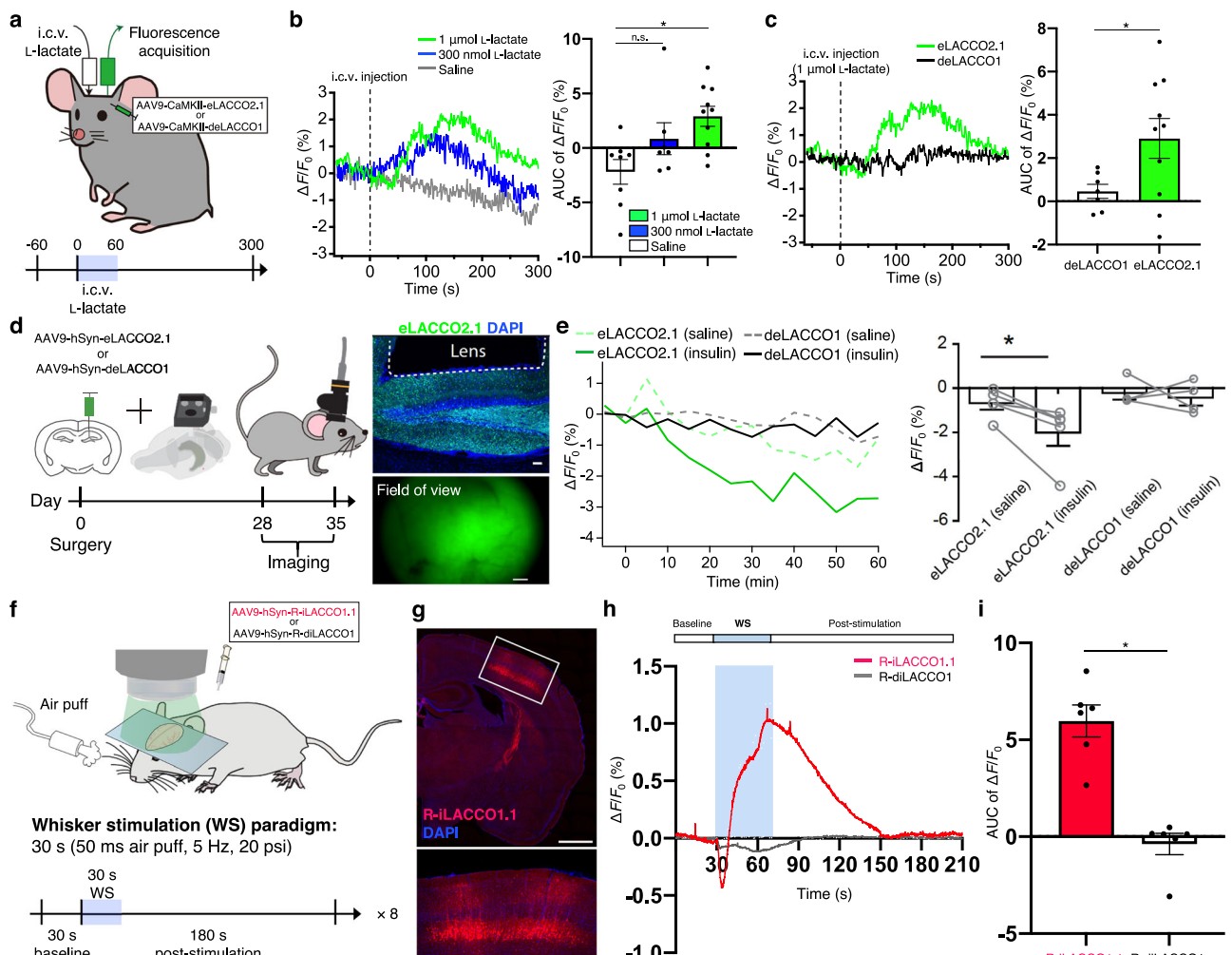

**Fig. 5 | In vivo L-lactate imaging in mice. a** Schematic illustration of in vivo mouse imaging upon i.c.v. L-lactate injection. **b** Fluorescence response traces (left) of the visual cortex in CaMKII-HA-eLACCO2.1-NGR expressing mice upon i.c.v. injection of L-lactate. Right panel shows the area under the curve (AUC) of the traces for each stimulation condition. Saline ($n = 8$ traces from three mice), 300 nmol L-lactate ($n = 7$ traces from three mice), and 1 µmol L-lactate ($n = 10$ traces from three mice). Mean ± s.e.m. One-way ANOVA with the Dunnett's post hoc tests. *$P = 0.0066$. **c** Fluorescence response traces (left) upon i.c.v. injection of 1 µmol L-lactate. Right panel shows the AUC of the traces for each construct. eLACCO2.1 ($n = 10$ traces from three mice) and deLACCO1 ($n = 7$ traces from three mice). Mean ± s.e.m. Two-tailed Student's $t$ test. *$P = 0.0493$. **d** Timeline of animal surgery and recording. The right panels show representative images of the coordinate of integrated GRIN lens implantation (upper) and eLACCO2.1 in the field of view (bottom). Scar bars, 50 µm (upper) and 100 µm (bottom). **e** Fluorescence responses in freely moving mice in

response to the i.p. injection of saline or insulin. Each recording was normalized to 10 min baseline recording. $n = 5$ and four mice for eLACCO2.1 and deLACCO1, respectively. Mean ± s.e.m. Two-tailed paired Student's $t$ test. *$P = 0.02533$. **f** Schematic illustration of in vivo one-photon imaging of the mouse stimulated by air pff. **g** Histological verification of R-iLACCO1.1 expression in the somatosensory cortex. Bottom image shows a magnified image in the box of the upper image. Similar results were obtained for more than five independent experiments. Scale bars, 1 mm (upper) and 200 µm (bottom). **h** Traces of air puff-evoked intracellular L-lactate response (mean ± s.e.m.). R-iLACCO1.1 ($n = 8$ traces from 6 mice of which one mouse was subjected to shorter post stimulation of 120 s) and R-diLACCO1 ($n = 8$ traces from 6 mice). **i** AUC of the traces for R-iLACCO1.1 ($n = 6$ mice) and R-diLACCO1 ($n = 6$ mice). Mean ± s.e.m. Two-tailed Student's $t$ test. *$P < 0.0001$. Source data are provided as a Source Data file.

concentration decreases (Fig. 6f, i). This finding suggests that transporters allowing L-lactate to travel across membranes exist on mitochondrial and ER membranes, similar to MCT on plasma membranes. This is consistent with previous reports that suggest that the lactate can shuttle between cytosol and mitochondria[39], and cytosol and ER[11]. Interestingly, L-lactate in mitochondria and ER might be exchanged within these organelles via membrane fusion (Fig. 6e, h). In analogy to calcium ion ($Ca^{2+}$) homeostasis between cellular compartments[40], the L-lactate concentration might be controlled by dynamic shuttling between cytosol, extracellular environment, mitochondria, and ER.

In summary, we have developed an improved green fluorescent extracellular L-lactate biosensor eLACCO2.1 and a red fluorescent intracellular L-lactate biosensor R-iLACCO1. This LACCO biosensor

series should provide new opportunities to investigate the emerging roles of L-lactate in spatially and spectrally multiplexed manner.

## Methods
### Ethical statement
For experiments performed at The University of Tokyo, all methods for animal care and use were approved by the institutional review committees of School of Science, The University of Tokyo. For experiments at HHMI Janelia Research Campus, all surgical and experimental procedures were in accordance with protocols approved by the HHMI Janelia Research Campus Institutional Animal Care and Use Committee and Institutional Biosafety Committee. For experiments at University of Copenhagen, the procedures involving animal care, surgery, in vivo imaging, and sample preparation were approved by the local research

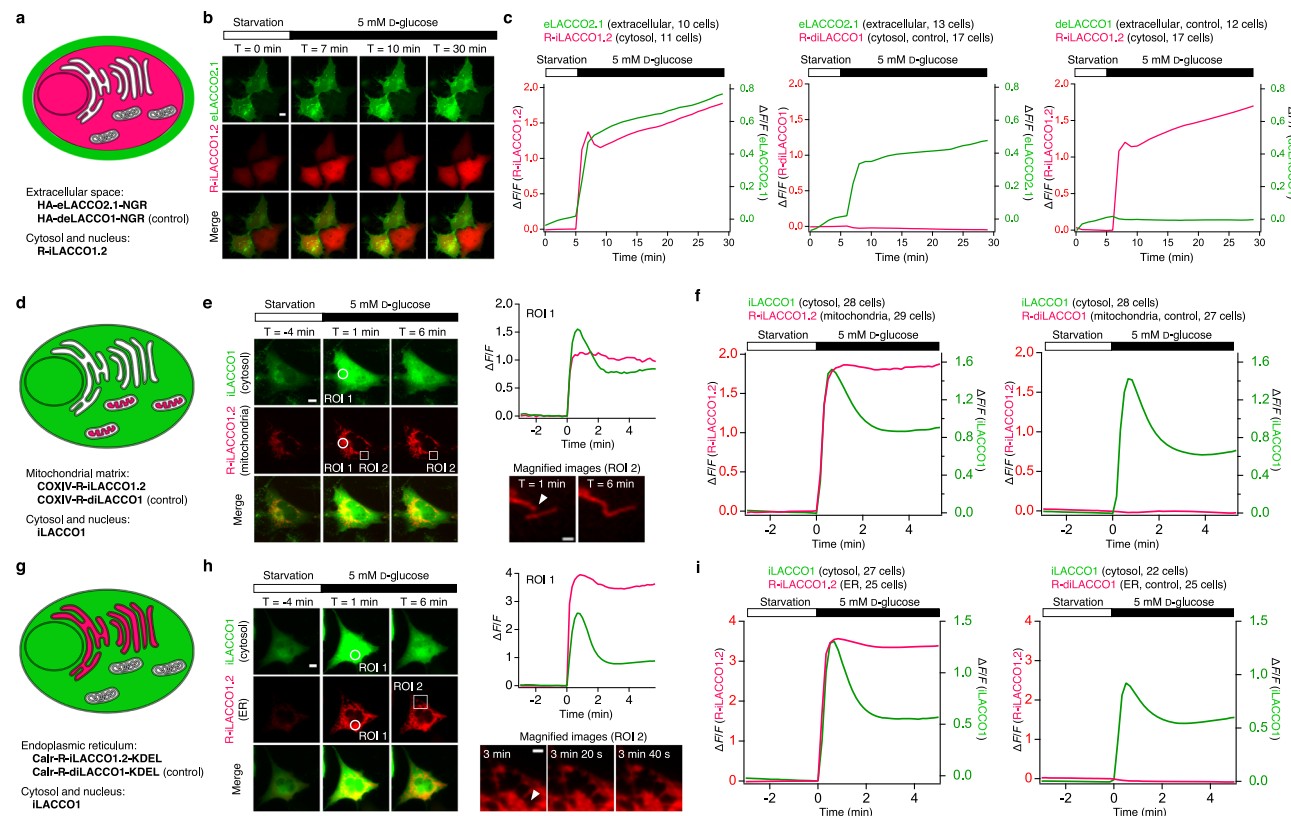

**Fig. 6 | Spectrally and spatially multiplexed ʟ-lactate imaging. a** Schematic illustration of ʟ-lactate imaging in extracellular space and cytosol. **b** Representative images of cell surface-targeted eLACCO2.1 and cytosolic R-iLACCO1.2 expressed in T98 cells before and after 5 mM glucose treatment. Similar results were observed in more than ten cells. Scale bar, 10 µm. **c** Fluorescence traces for imaging of extracellular and cytosolic ʟ-lactate in T98 cells upon glucose treatment. Mean ± s.e.m. **d** Schematic illustration of ʟ-lactate imaging in mitochondrial matrix and cytosol. COXIV N-terminal yeast cytochrome *c* oxidase subunit IV (MLSLRQSIRFFKRSGI). **e** Representative images of cytosolic iLACCO1 and mitochondrial matrix-targeted R-iLACCO1.2 expressed in T98 cells before and after 5 mM glucose treatment. Similar results were observed in more than ten cells. White arrowhead indicates a

mitochondrial fusion. Scale bars, 10 µm and 2 µm (magnified image). **f** Fluorescence traces for imaging of ʟ-lactate in cytosol and mitochondrial matrix in T98 cells upon glucose treatment. Mean ± s.e.m. **g** Schematic illustration of ʟ-lactate imaging in endoplasmic reticulum (ER) and cytosol. Calr N-terminal human calreticulin (MLLPVLLLGLLGAAAD). **h** Representative images of cytosolic iLACCO1 and ER-targeted R-iLACCO1.2 expressed in T98 cells before and after 5 mM glucose treatment. Similar results were observed in more than ten cells. White arrowhead indicates an ER fusion. Scale bars, 10 µm and 3 µm (magnified image). **i** Fluorescence traces for imaging of ʟ-lactate in cytosol and ER in T98 cells upon glucose treatment. Mean ± s.e.m. Source data are provided as a Source Data file.

ethics committee (Department of Experimental Medicine, University of Copenhagen) and conducted in accordance with the Danish Animal Experiments Inspectorate. All animal experiments at Kagoshima University were approved by the Experimental Animal Research Committee of Kagoshima University (Approval number: MD22086) and the Gene Recombination Experiment Safety Committee of Kagoshima University (Approval numbers: 22S018), and were performed in accordance with the ARRIVE guidelines, the National Institutes of Health (NIH) guide for the care and use of laboratory animals, and the Code of Ethics of the World Medical Association for animal experiments (http://ec.europa.eu/environment/chemicals/lab_animals/legislation_en.htm). All surgeries and experimental procedures at Stony Brook University were reviewed and approved by the Stony Brook University Animal Use Committee and followed the guidelines of NIH. The Stony Brook University IACUC-monitored Division of Laboratory Animal Resources maintained all animals. For experiments performed at University of Calgary, all methods for animal care and use were approved by the University of Calgary Animal Care and Use Committee and were in accordance with the National Institutes of Health Guide for the Care and Use of Laboratory Animals.

### General methods and materials
Synthetic DNA encoding LldR, CanlonicSF, Green Lindoblum, GEM-IL ver.3, and FiLa were purchased from Integrated DNA Technologies. A

gene encoding Laconic was obtained from Addgene (Addgene plasmid #44238; http://n2t.net/addgene:44238). Phusion high-fidelity DNA polymerase (Thermo Fisher Scientific) was used for routine polymerase chain reaction (PCR) amplifications, and Taq DNA polymerase (New England Biolabs) was used for error-prone PCR. Restriction endonucleases, rapid DNA ligation kits, and GeneJET miniprep kits were purchased from Thermo Fisher Scientific. PCR products and products of restriction digests were purified using agarose gel electrophoresis and the GeneJET gel extraction kit (Thermo Fisher Scientific). DNA sequences were analyzed by DNA sequence service of Fasmac Co., Ltd. Fluorescence excitation and emission spectra were recorded on a Spark plate reader (Tecan).

### Structural modeling of R-iLACCO1
The modeling structure of R-iLACCO1 was generated by AlphaFold2 (refs. [41],[42]) using an API hosted at the Söding lab in which the MMseqs2 server[43] was used for multiple sequence alignment.

### Engineering of eLACCO2.1 and R-iLACCO1
A previously reported green extracellular ʟ-lactate biosensor intermediate eLACCO0.9 (ref. [9]) was subjected to an iterative process of library generation and screening in *E. coli* strain DH10B (Thermo Fisher Scientific) in LB media supplemented with 100 µg mL⁻¹ ampicillin and 0.02% ʟ-arabinose. Libraries were generated by error-prone PCR of the

whole gene. Two rounds of the directed evolution led to the identification of eLACCO1.2. Cys340Ser mutation was introduced into eLACCO1.2. The resulting variant, designated eLACCO1.3, was subjected to an iterative process of library generation and screening in *E. coli* by error-prone PCR of the whole gene. For each round, ~200–400 fluorescent colonies were picked, cultured, and tested on 96-well plates under a plate reader. There were 6 rounds of screening before eLACCO2 was identified. Finally, Leu79Ile mutation was added to eLACCO2 to tune the L-lactate affinity. The resulting mutant was designated as eLACCO2.1. Asp444Asn mutation was added to eLACCO2 to abrogate the L-lactate binding. Three rounds of directed evolution were performed to improve the brightness. The resulting control mutant was designated as deLACCO1.

The gene encoding cpmApple with N- and C- terminal linkers (DW and EREG, respectively) was amplified using a red extracellular L-lactate biosensor R-eLACCO1 gene as template[18], followed by insertion into the site between Leu109 and Val110 of LldR L-lactate binding protein in a pBAD (Invitrogen) by Gibson assembly (New England Biolabs). The resulting variant was designated as R-iLACCO0.1. N- and C-terminal linkers of R-iLACCO0.1 were deleted using Q5 high-fidelity DNA polymerase (New England Biolabs) to provide variants with different linker length. Variants were expressed in *E. coli*. Proteins were extracted using B-PER bacterial protein extraction reagent (Thermo Fisher Scientific) and tested for fluorescence brightness and L-lactate-dependent response. The most promising variant, designated as R-iLACCO0.2, was subjected to an iterative process of library generation and screening in *E. coli*. Libraries were generated by site-directed random mutagenesis or error-prone PCR of the whole gene. For each round, approximately 200–400 fluorescent colonies were picked, cultured, and tested on 96-well plates under a plate reader. There were 11 rounds of screening before R-iLACCO1, R-iLACCO1.1, and R-iLACCO1.2 were identified. Asp69Asn mutation was added to R-iLACCO1 to abrogate the L-lactate affinity. Two rounds of directed evolution were performed to improve the brightness. The resulting control mutant was designated as R-diLACCO1.

### Protein purification and in vitro characterization

The gene encoding eLACCO2.1 and R-iLACCO variants, with a poly-histidine tag on the N-terminus, was expressed from the pBAD vector. Bacteria were lysed with a cell disruptor (Branson) and then centrifuged at 15,000 $g$ for 20 min, and proteins were purified by Ni-NTA affinity chromatography (Agarose Bead Technologies). Absorption spectra of the samples were collected with a Lambda950 Spectrophotometer (PerkinElmer). To perform pH titrations for eLACCO2.1, protein solutions were diluted into buffers (pH from 4 to 11) containing 30 mM trisodium citrate, 30 mM sodium borate, 30 mM MOPS, 100 mM KCl, 1 mM CaCl₂, and either no L-lactate or 100 mM L-lactate. To perform pH titrations for R-iLACCO variants, protein solutions were diluted into buffers (pH from 4 to 11) containing 30 mM trisodium citrate, 30 mM sodium borate, 30 mM MOPS, 100 mM KCl, and either no L-lactate or 100 mM L-lactate. Fluorescence intensities as a function of pH were then fitted by a sigmoidal binding function to determine the p$K_a$. For L-lactate titration of eLACCO2.1, buffers were prepared by mixing an L-lactate (−) buffer (30 mM MOPS, 100 mM KCl, 1 mM CaCl₂, pH 7.2) and an L-lactate (+) buffer (30 mM MOPS, 100 mM KCl, 1 mM CaCl₂, 100 mM L-lactate, pH 7.2) to provide L-lactate concentrations ranging from 0 mM to 100 mM at 25 °C. For L-lactate titration of R-iLACCO variants, buffers were prepared by mixing an L-lactate (−, w/o Ca²⁺) buffer (30 mM MOPS, 100 mM KCl, pH 7.2) and an L-lactate (+, w/o Ca²⁺) buffer (30 mM MOPS, 100 mM KCl, 100 mM L-lactate, pH 7.2) to provide L-lactate concentrations ranging from 0 mM to 100 mM at 25 °C. Fluorescence intensities were plotted against L-lactate concentrations and fitted by a sigmoidal binding function to determine the Hill coefficient and apparent $K_d$. For Ca²⁺ titration, buffers were prepared by mixing a Ca²⁺ (−) buffer (30 mM MOPS, 100 mM KCl,

10 mM EGTA, 100 mM L-lactate, pH 7.2) and a Ca²⁺ (+) buffer (30 mM MOPS, 100 mM KCl, 10 mM CaEGTA, 100 mM L-lactate, pH 7.2) to provide Ca²⁺ concentrations ranging from 0 µM to 39 µM at 25 °C. Buffers with Ca²⁺ concentrations more than 39 µM was prepared by mixing a Ca²⁺ (−, w/o EGTA) buffer (30 mM MOPS, 100 mM KCl, 100 mM L-lactate, pH 7.2) and a Ca²⁺ (+, w/o EGTA) buffer (30 mM MOPS, 100 mM KCl, 100 mM CaCl₂, 100 mM L-lactate, pH 7.2).

Two-photon excitation spectra and two-photon absorption cross sections of eLACCO2.1 and R-iLACCO1 were measured according to the general methods and protocols described in ref. 44. Briefly, a tunable femtosecond laser InSight DeepSee (Spectra-Physics, Santa Clara, CA) was used to excite fluorescence of the sample contained within a PC1 Spectrofluorometer (ISS, Champaign, IL). To measure the two-photon excitation spectral shapes, we used short-pass filters 633SP and 770SP in the emission channel. LDS 798 in 1:2 CHCl3:CDCl3 and Coumarin 540 A in DMSO were used as spectral standards. Quadratic power dependence of fluorescence intensity in the proteins and standards was checked at several wavelengths across the spectrum. The two-photon cross section ($\sigma_2$) of the anionic form of the chromophore in eLACCO2.1 and R-iLACCO1 was measured as described in ref. 45. Rhodamine 6 G in methanol was used as a reference standard with excitation at 1060 nm (ref. 44). For one-photon excitation, we used a 532-nm line of diode laser (Coherent). A combination of filters 770SP and 561LP was used in the fluorescence channel for these measurements. Extinction coefficients were determined by alkaline denaturation as detailed in ref. 46. The two-photon absorption spectra were normalized to the measured $\sigma_2$ values. To normalize to the total two-photon brightness ($F_2$), the spectra were then multiplied by the quantum yield and the relative fraction of the respective form of the chromophore for which the $\sigma_2$ was measured. The data is presented this way because eLACCO2.1 and R-iLACCO1 contain a mixture of the neutral and anionic forms of the chromophore. The method is described in further detail in refs. 46,47.

### Construction of mammalian expression vectors

For cell surface expression, the genes encoding eLACCO2.1 were amplified by PCR followed by digestion with BglII and EcoRI, and then ligated into a pcDNA3.1 vector (Thermo Fisher Scientific) that contains N-terminal leader sequence and C-terminal anchor. N-terminal leader sequences were derived from ref. 18 and C-terminal anchors stemmed from refs. 18,20. To construct R-iLACCO variants for intracellular expression, the gene encoding R-iLACCO variants in the pBAD vector was digested with XhoI and HindIII, and then ligated into pcDNA3.1 vector. To construct plasmids for expression in primary neurons, astrocytes, ex vivo, and in vivo, the gene encoding R-iLACCO variants or eLACCO variants including the leader and anchor sequence in the pcDNA3.1 vector was first amplified by PCR followed by digestion with NheI and XhoI for eLACCO and BamHI and HindIII for R-iLACCO, and then ligated into a pAAV plasmid containing the human synapsin (hSyn) or gfaABC1D promoter. To construct plasmids for expression in principal cortical neurons in vivo, the gene encoding eLACCO variants including the leader and anchor sequence in the pcDNA3.1 vector was first amplified by PCR followed by digestion with BamHI and HindIII, and then ligated into a pAAV plasmid containing the human Ca²⁺/calmodulin-dependent protein kinase II α (CaMKIIα, 1.3 kbp) promoter.

### Imaging of eLACCO and R-iLACCO variants in HeLa, HEK293T, and T98G cell lines

HeLa (American Type Culture Collection, ATCC, #CCL-2) and HEK293T (Thermo Fisher Scientific, #R70007) cells were maintained in Dulbecco's modified Eagle medium (DMEM; Nakalai Tesque) supplemented with 10% fetal bovine serum (FBS; Sigma-Aldrich) and 1% penicillin–streptomycin (Nakalai Tesque) at 37 °C and 5% CO₂. T98G cells (ATCC, #CRL-1690) were maintained in minimum essential

medium (Nakalai Tesque) supplemented with 10% FBS, 1% penicillin–streptomycin, 1% non-essential amino acid (Nakalai Tesque) and 1 mM sodium pyruvate (Nakalai Tesque) at 37 °C and 5% $CO_2$. Cells were seeded in 35-mm glass-bottom cell-culture dishes (Iwaki) and transiently transfected with the constructed plasmid using poly-ethyleneimine (Polysciences). Transfected cells were imaged using a IX83 wide-field fluorescence microscopy (Olympus) equipped with a pE-300 LED light source (CoolLED), a 40× objective lens (numerical aperture (NA) = 1.3; oil), an ImagEM X2 EM-CCD camera (Hamamatsu), Cellsens software (Olympus), and an STR stage incubator (Tokai Hit). The filter sets used in live cell imaging had the following specification. eLACCO variants, iLACCO1, CanlonicSF, Green Lindoblum, FiLa (ex. 470 nm), and EGFP: excitation 470/20 nm, dichroic mirror 490-nm dclp, and emission 518/45 nm; R-iLACCO variants, mRFP1, and mApple: excitation 545/20 nm, dichroic mirror 565-nm dclp, and emission 598/55 nm; GEM-IL and Laconic (CFP): excitation 438/24 nm, dichroic mirror 458-nm dclp, and emission 483/32 nm; Laconic (FRET): excitation 438/24 nm, dichroic mirror 458-nm dclp, and emission 542/27 nm; FiLa (ex. 405 nm): excitation 405/20 nm, dichroic mirror 425-nm dclp, and emission 518/45 nm. Fluorescence images were analyzed with ImageJ software (National Institutes of Health).

For photostability test, HeLa cells transfected with respective genes were illuminated by excitation light at 100% intensity of LED (~10 mW cm$^{-2}$ and ~4 mW cm$^{-2}$ on the objective lens for eLACCO and R-iLACCO variants, respectively) and their fluorescence images were recorded for 4 min with the exposure time of 50 ms and no interval time.

For imaging of L-lactate-dependent fluorescence changes, HeLa cells seeded onto coverslips were transfected with plasmids encoding eLACCO or R-iLACCO variants. Forty-eight hours after transfection, the coverslips were transferred into Attofluor™ Cell Chamber (Thermo Fisher Scientific) with Hank's balanced salt solution (HBSS(+); Nakalai Tesque) supplemented with 10 mM HEPES and 1 μM AR-C155858 (Tocris). Exchange of bath solutions during the image was performed in a remove-and-add manner using a homemade solution remover[9].

For imaging of $Ca^{2+}$-dependent fluorescence, HeLa cells on coverslips were transferred into Attofluor™ Cell Chamber with HBSS(−) buffer (Nakalai Tesque) supplemented with 10 mM HEPES and 10 mM L-lactate, 48 h after transfection of eLACCO variants. Other bath solutions were supplemented with $Ca^{2+}$ of 100 nM, 1 μM, 10 μM, 100 μM, 300 μM, 1 mM, 3 mM, and 10 mM. Exchange of bath solutions during the image was performed in the same way as the imaging of L-lactate-dependent fluorescent changes.

## Imaging of eLACCO and R-iLACCO variants in primary neurons and astrocytes

The neuron imaging was previously described[9]. In short, rat cortical/hippocampal primary cultures from the P0 pups (pooled tissues from males and females) were plated in glass-bottom 24-well plates where $0.5 \times 10^6$ cells were used for three wells. Cultures were nucleofected at time of plating, and imaged 14 days later. For the imaging of primary astrocytes, male and female pups were obtained from a single timed-pregnant Sprague Dawley rat (Charles River Laboratories, purchased from Japan SLC, Inc.). Experiments were performed with cortical/hippocampal primary cultures from the E21 (after C-section of the pregnant rat) plated in glass-bottom 24-well plates (Cellvis) where $0.5 \times 10^6$ cells were used for one well. Cultures were nucleofected at time of plating with Nucleofector 4D (Lonza), and imaged 5 days later. Astrocytes were cultured in DMEM supplemented with 10% FBS and 1% penicillin–streptomycin at 37 °C and 5% $CO_2$. Culture media were replaced with 1 mL of imaging buffer (145 mM NaCl, 2.5 mM KCl, 10 mM D-glucose, 10 mM HEPES, 2 mM $CaCl_2$, 1 mM $MgCl_2$, pH 7.4) for imaging.

## Imaging of R-iLACCO1.2 in myotubes

Mouse C2C12 myoblasts (#CRL-1772, ATCC) were grown in DMEM supplemented with 10% FBS before induction of differentiation. After reaching confluence 2 days post-seeding, the culture medium was replaced with a differentiation medium (DMEM supplemented with 1% horse serum (Gibco)), and the medium was changed every 2 days until 6 days after differentiation. Retroviruses were used to express R-iLACCO1.2 and R-diLACCO1 in C2C12 myotubes. To construct retrovirus vectors, the genes encoding R-iLACCO1.2 and R-diLACCO1 were amplified by PCR followed by digestion with BamHI and EcoRI and then ligated into a pMX vector. Retrovirus was produced using plat-E packaging cells (Cell Biolabs) which were transfected with pMX plasmids using TransIT-LT1 reagent (TakaraBio). Retrovirus-containing supernatants were collected 60 h after transfection. To transduce retrovirus, undifferentiated C2C12 myoblasts were plated in a 35-mm dish at 40% confluency. Retrovirus was added to the medium three times at intervals of 6 h, and the medium was changed to culture medium 12 h after the final infection. For the observation of glucose-dependent fluorescence in myotubes, undifferentiated myoblasts expressing R-iLACCO1.2 or R-diLACCO1 were seeded to eight-well μ-Slide (ibidi) and differentiated into myotubes. Culture media was changed to imaging buffer without glucose (Krebs-Ringer phosphate buffer; 5 mM $KH_2PO_4$, 136 mM NaCl, 4.7 mM KCl, 1 mM $CaCl_2$, 1 mM $MgSO_4$, 25 mM $NaHCO_3$, 20 mM HEPES) 2 h before observation. Myotubes were imaged using an FV-1200 laser-scanning confocal fluorescence microscope (Olympus) equipped with a UPLXAPO 10× air/dry objective (NA 0.40), and 559-nm laser for R-iLACCO1.2 and R-diLACCO1 excitation. Myotubes were maintained at 37 °C with a stage top incubator (STX, Tokai Hit).

## Clustering analysis

Clustering analyses were performed using R (R version 4.2.3; R Core Team). For clustering analysis, the traces of $\Delta F/F$ from glucose-stimulated cells were used. Among 42 cells analyzed, 21 cells were pretreated with MCT inhibitor, AR-C155858. To group cells that showed similar fluorescence traces upon glucose addition, hierarchical clustering was performed using Euclidean distance by Ward's method. The resulting dendrogram and a heatmap were generated using the R package "ggplot2".

## Transgenic line generation in *Drosophila melanogaster*

The coding sequence of the corresponding biosensor was amplified from pcDNA3.1 by PCR and subsequently inserted into EcoRI/XbaI site of pUAST-attB vector using In-Fusion Snap Assembly Master Mix (Takara). The transgenic lines were generated via Φ integrase-mediated recombination into the fly genome at landing site attp2 (Rainbow Transgenic Flies Inc).

## Live imaging of ex vivo *Drosophila* adult brains

Live imaging of ex vivo *Drosophila* adult brains (3–5 day old females) were performed at room temperature with LSM800 confocal microscope (Zeiss). Whole brain images were captured with a 20× Plan-Apochromat objective (NA 0.8) in single plane mode (scan speed 8; 1024 × 1024-pixel; 8 bits per pixel; averaging number 16; pinhole 1–1.5 AU). Time series images were taken every 60 s for 5 min with a 63× Plan-Apochromat Oil DIC objective (NA 1.4) in z-stack mode (scan speed 8; 1024 × 1024-pixel; 8 bits per pixel; averaging number 4; pinhole 1–1.5 AU; z-stack 5 slices). Adult brains were quickly dissected in Schneider's Drosophila Medium (Gibco) and then put on a poly-lysine coated glass bottom dish (Matsunami Glass Ind. Ltd.) filled with HL3 buffer (70 mM NaCl, 5 mM KCl, 20 mM $MgCl_2$, 10 mM $NaHCO_3$, 115 mM sucrose, 5 mM HEPES; pH 7.2) supplemented with 5 mM D-glucose, 1 mM L-lactate, and 0.5 mM pyruvate[11]. The dish was placed on the microscope stage, and the medium in the dish was replaced with an HL3 buffer containing 6 mM oxalate. After 5 min, L-lactate was added to a final concentration of 10 mM. Samples were excited at 561 nm, and the emission was recorded at 566–700 nm.

## Packaging and purification of adeno-associated viruses (AAVs)

AAVs were generated in HEK293T17 cells by a triple transfection of a helper plasmid pXX680, a plasmid encoding either the AAV2/5 or AAV2/9 rep/cap, and a pAAV plasmid encoding the corresponding LACCO biosensor. Forty-eight hours post transfection, the cells were harvested by gentle scraping and pelleted by centrifugation at $2300 \times g$. Viral particles were released by four freeze-thaw cycles on dry-ice/ethanol and free DNA was digested with benzonase. The AAV particles were purified by a discontinuous gradient of iodixanol (15-25-40-60%) and ultracentrifugation ($462,000 \times g$ for 70 min at 16 °C). The viral preparation was then washed and concentrated through a 100 kD amicon filter. During the last concentration step, the buffer was adjusted to 5% D-sorbitol and 0.001% pluronic acid for storage and experimentation. Titration was performed by Taq-Man digital droplet-PCR using primers specific for AAV2 inverted terminal repeat (ITR).

## Preparation of mouse acute cortical brain slice

At 4 weeks after the injection of AAV encoding hSyn-HA-eLACCO2.1-NGR or gfaABC1D-HA-eLACCO2.1-NGR, mice (C57Bl6, male, P28, housed at 21 °C with 47% humidity) were anaesthetized with gaseous isoflurane (5%) and then decapitated. The brain was removed, then submerged for 2 min in ice-cold slicing solution containing: 119.9 mM N-methyl-D-glucamine, 2.5 mM KCl, 25 mM NaHCO₃, 1.0 mM CaCl₂·2H₂O, 6.9 mM MgCl₂·6H₂O, 1.4 mM NaH₂PO₄·H₂O, and 20 mM D-glucose. The brain was then Krazy Glued onto a vibratome tray (Leica Instruments, VT1200S) and then resubmerged in ice-cold slicing solution. Acute coronal slices were prepared from the somatosensory cortex (400 μm thick) using a vibratome. The slices were incubated for 45 min at 33 °C in a recovery chamber filled with artificial cerebrospinal fluid (ACSF) containing: 126 mM NaCl, 2.5 mM KCl, 25 mM NaHCO₃, 1.3 mM CaCl₂·2H₂O, 1.2 mM MgCl₂·6H₂O, 1.25 mM NaH₂PO₄·H₂O, and 10 mM D-glucose. After recovery, slices were stained with astrocyte marker sulforhodamine 101 (SR101, 10 μM for 20 min, recovery for 1 h) to visualize astrocyte processes to maintain focal plane during the imaging experiments. Throughout, brain slices were continuously supplied with carbogen (95% oxygen, 5% CO₂).

## Two-photon microscopy of eLACCO2.1 in acute cortical brain slice

Brain slices were imaged using a custom built two-photon microscope fed by a Ti:Sapphire laser source (Coherent Ultra II, ~4 W average output at 800 nm, ~80 MHz). Image data were acquired using MATLAB (2013) running the open source scanning microscope control software ScanImage (version 3.81, HHMI/Janelia Research Campus). Imaging was performed at an excitation wavelength of 940 nm. The microscope was equipped with a primary dichroic mirror at 695 nm and green and red fluorescence was split and filtered using a secondary dichroic at 560 nm and two bandpass emission filters: 525–540 nm and 605–670 nm (Chroma Technologies). Time series images were acquired at 0.98 Hz with a pixel density of 512 by 512 and a field of view size of ~292 μm. Imaging used a ×40 water dipping objective lens (NA 1.0, WD 2.5 mm, Zeiss). Imaging was performed at room temperature.

For aglycemia experiments in acute brain slices, glucose was not added to the ACSF and in its place 5 mM extra NaCl was supplemented to maintain osmolarity (10 mosmols). For electrical afferent stimulation in acute slices, a Grass S88X stimulator, a voltage isolation unit, and a concentric bipolar electrode (FHC) assembly were used. The electrode was positioned onto the surface of the slice by a micromanipulator ~300 μm away from the region of interest. The tissue was stimulated with 1 ms long mono-phasic pulses at 1.2 V using a theta burst pattern (100 Hz for 50 ms, repeated at 4 Hz). Theta burst was applied for 5, 15 and 30 s in separate experiments. The different lengths of stimulation produced similar results and the data was pooled.

## In vivo validation of eLACCO2.1

Mice (C57BL6/J, male, 8–9 weeks, housed at 23 °C with 40–70% humidity) were anesthetized by intraperitoneal (i.p.) injection of the mixture consisting of 0.75 mg kg⁻¹ medetomidine (Domitor® Nippon Zenyaku Kogyo Co., Ltd., Tokyo, Japan), 4.0 mg kg⁻¹ midazolam (Dormicum®, Astellas Pharma Inc., Tokyo, Japan), and 5.0 mg kg⁻¹ butorphanol (Vetorphale®, Meiji Seika Pharma, Co., Ltd., Tokyo, Japan). Holes were drilled on the skull. Glass needles (Ringcaps, Hirschmann Laborgeräte GmbH & Co. Eberstadt, Germany) were pulled by a puller (PE-22, Narishige, Tokyo, Japan) and stereotaxically introduced into the visual cortex. Half mL of AAV9-CaMKIIα-HA-eLACCO2.1-NGR or AAV9-CaMKIIα-HA-deLACCO1-NGR ($4 \times 10^{13}$ viral genome (vg) mL⁻¹ each) was unilaterally injected, and needles were settled at the injected places for 10 min to defuse the viral vectors. After the viral injection, a fiber optic cannula (Lucir, Tsukuba, Japan) was implanted into the same visual cortex. For the intracerebroventricular (i.c.v.) injection, a guide cannula (Eicom, Kyoto, Japan) was placed into the lateral ventricle. The coordinates of specific brain regions were as follows: the visual cortex (lateral, +2.7 mm; posterior, −2.8 mm from the bregma and ventral, −0.4 mm from the brain surface), and the lateral ventricle (lateral, −1.0 mm; posterior, −0.2 mm from the bregma and ventral, −1.8 mm from the brain surface) according to the mouse brain atlas. A fiber optic cannula and an injection cannula were fixed on the mice's skulls using dental composite resin. Subsequently, mice were injected with 0.75 mg kg⁻¹ atipamezole (i.p., FUJIFILM Wako Pure Chemical Corporation, Osaka, Japan) to awaken them and 0.8 mg per mouse of penicillin G (s.c., FUJIFILM Wako Pure Chemical Corporation) to prevent infections. Three weeks later, the mice were subjected to i.c.v. L-lactate injection.

On the day of the experiment, an optical fiber was connected to a fiber optical cannula, and an injection cannula was inserted into a guide cannula. L-Lactate (300 nmol and 1 μmol) or saline was administered intracerebroventricularly via an injection cannula (1 mL min⁻¹). eLACCO fluorescence was acquired by the fiber photometry system (COME2-FTR/GFP, Lucir, Tsukuba, Japan). A fiber-coupled LED (LEDD1B/M470F3, Thorlabs, Newton, NJ) produced excitation blue light (470 nm) of power 0.1 mW at the tip of the silica fiber; the blue light to pass through the excitation bandpass filter (passband 475 ± 12.5 nm), was reflected by a dichroic mirror-1, and joined the single silica fiber (diameter 400 μm; NA 0.6). The blue light emitted from the tip of the fiber optical cannula reflected the eLACCO fluorescent proteins; the reflected green fluorescence signals were transmitted to the same silica fiber. The signals to pass through the dichroic mirror, were reflected by dichroic mirror-2, and passed through the bandpass emission filter (passband: 510 ± 12.5 nm). Finally, the signals were guided to a photomultiplier (PMTH-S1-1P28, Zolix Instruments, Beijing, China). The signals were digitized by an A/D converter (PowerLab2/25, AD Instruments Inc., Dunedin, New Zealand) and recorded with the LabChart version-7 software (AD Instruments Inc.). Data was acquired at 1 Hz. Baseline fluorescence intensity ($F_0$) was calculated as the average of the 10 s before L-lactate administration, and fluorescence intensity ($F$) at each time point was used to calculate $\Delta F/F = (F-F_0)/F_0$. The area under the curve (AUC) was calculated by adding the $\Delta F/F$ over 5 min after L-lactate or saline administration.

## In vivo eLACCO2.1 imaging

All mice were housed in a 12-h light/dark cycle condition at 18–26 °C with 30–70% humidity. All behavioral experiments were performed during the light cycle. Mice were provided ad libitum access to food and water. Surgeries were conducted using 8–10 week-old male mice (C57BL/6, Charles River Laboratories). Mice were anesthetized with isoflurane at 4% for induction and 1.5–2% during surgery with 0.4 L min⁻¹ O₂, while the depth of anesthesia was monitored by breathing rate. AAV9-hSyn-HA-eLACCO2.1-NGR or AAV9-hSyn-HA-deLACCO1-NGR was stereotaxically injected into the dorsal dentate

**Article**

gyrus (AP: −2.0, ML: 1.5, DV: 2.5). After about 1 h after viral injection, an integrated GRIN lens (outer diameter: 1.0 mm, length: 4.0 mm) was implanted approximately 200 mm above the dentate granule cell (DGC) layer. Mice were allowed to have 4 weeks for viral expression and recovery from surgery. All in vivo imaging videos were recorded from freely moving animals (4–5 weeks after surgery) in their home cage by using a miniature microscope and the nVue data acquisition system (Inscopix, Palo Alto, CA) at a frame rate of 20 Hz. For each recording, animals were allowed 10 min to familiarize the recording setup, and a 10 min baseline recording was performed before insulin treatment (i.p. injection, 0.75 U kg$^{-1}$). The same volume of saline injection with some recording paradigm was set as a control experiment. The order for treatment and control recordings was randomly assigned. The fluorescence intensity of the region of interest of images was analyzed in the Inscopix Imaging Processing (ver. 1.8.0). MATLAB (R2022) was used for further analysis and the final data plot.

## In vivo R-iLACCO imaging
Mice (C57BL/6 J, male, 8–15 weeks, housed at 21 °C with 40–60% humidity) were anesthetized by 3–4% isoflurane for induction and the anesthesia was maintained at 1–1.5% isoflurane while they were mounted to the stereotaxic frame. The body temperature was maintained at 37 °C throughout the surgery. The skull was exposed after applying local analgesia (lidocaine, 0.2 mg mL$^{-1}$) by making an incision to the scalp, and a metal frame (head plate) was then attached to the skull using dental cement (Super Bond C&B, Sun Medical, Shiga, Japan). A 4-mm diameter craniotomy above the somatosensory cortex was made and the dura mater was surgically removed. AAVs were injected into the somatosensory cortex at depths of 300 μm (1.3–1.5 × 10$^{12}$ vg mL$^{-1}$ in PBS, 500 nL) using a glass pipette attached to a micromanipulator and a microinjector (FemtoJet, Eppendorf). A 4-mm diameter autoclaved cover slip was carefully mounted to cover the brain and then sealed by dental cement. Mice received a subcutaneous administration of carprofen (5 mg kg$^{-1}$) for systemic analgesia after the surgery and were recovered for at least three weeks in their home cage. To probe the performance of L-lactate biosensors in response to sensory stimuli, shallowly anesthetized (70 mg kg$^{-1}$ ketamine + 10 mg kg$^{-1}$ xylazine) mice were rigidly fixed under the fluorescence macroscope (self built, Thorlabs). Sample was excited to collect the fluorescence using an LED (pE4000, CoolLED), a 488/561 dual band filter set (59904, Chroma), and an EM-CCD camera (Andor iXon 888, frame rate 10 Hz). For sensory stimulus, air puffs (50 ms, 20 psi, 5 Hz, Picospritzer, Parker) were presented to the mouse's whisker pad for 30 s. The sensory stimulation session was repeated for 8 times including 30 s baseline and 120–180 s post-stimulation periods.

## Statistics and reproducibility
All data are expressed as mean ± s.d. or mean ± s.e.m., as specified in figure legends. Box plots are used for Figs. 3c, f, g, j, k, m, n and 4b, d–j. In these box plots, the horizontal line is the median; the top and bottom horizontal lines are the 25th and 75th percentiles for the data; and the whiskers extend one standard deviation range from the mean represented as black filled circle. Sample sizes (*n*) are listed with each experiment. No samples were excluded from analysis and all experiments were reproducible. In the representative fluorescence images in Figs. 3a, b, h, i and 6b, e, h, similar results were observed in more than five cells. Statistical analysis was performed using two-tailed Student's *t* test or one-way analysis of variance (ANOVA) with the Dunnett's post hoc tests (GraphPad Prism 9), as specified in figure legends. Microsoft Excel software was used to plot for Fig. 1d, e, g, h.

## Reporting summary
Further information on research design is available in the Nature Portfolio Reporting Summary linked to this article.

## Data availability
The plasmids generated in this study have been deposited in Addgene (https://www.addgene.org/depositing/83199/). Source data are provided with this paper.

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

## Acknowledgements

The authors thank T. Inui, S. Hario, T. Terai, Y. Shen, and M. Sato for technical support. Work at the University of Tokyo was supported by the Japan Society for the Promotion of Science (Grants-in-Aid for Early-Career Scientists 19K15691 (Y.N.), 21K14738 (Y.N.), and 22K14779 (G.K.), Grants-in-Aid for Transformative Research Areas A 23H04151 (Y.N.), and Grants-in-Aid for Scientific Research S 19H05633 (R.E.C.)), the Japan Science and Technology Agency PRESTO JPMJPR22E9 (Y.N.), Toyota Physical and Chemical Research Institute (Y.N.), The Precise Measurement Technology Promotion Foundation (Y.N.), and Suntory Foundation for Life Sciences (Y.N.). Work at Montana State University was supported by NIH grants U01 NS094246 (M.D.), U24 NS109107 (M.D.), and F31 NS108593 (M.D.). Work at University of Tsukuba was supported by the Japan Agency for Medical Research and Development JP21zf0127005 (H.T.). Work at University of Copenhagen and University of Rochester Medical Center were supported by the NIH grant U19NS128613 (M.N., H.H.), Danmarks Frie Forskningsfond 0134-00107B (H.H.), Novo Nordisk Foundation NNF19OC0058058 (H.H.), and Lundbeck Foundation R360-2021-613 (H.H.).

## Author contributions

Y.N. developed eLACCO2.1 and performed in vitro characterization. Y.N. and G.L. developed R-iLACCO variants. Y.N. performed in vitro characterization of R-iLACCO variants. Y.N., Y.K., A.A., and K.P. performed screening of leader and anchor sequences in primary neurons. M.D. measured one-photon absorbance spectra and two-photon excitation spectra. G.K. and T.O. performed myotube imaging and clustering analysis. H.T. developed transgenic flies and S.N. and Y.N. observed them. M.B. and M.-E.P produced AAVs. T.R.R., F.W., K.A.G., and G.R.G. performed mouse acute brain slice imaging. Y. Kambe produced CaMKII AAVs and performed in vivo validation of eLACCO2.1. X.W. and S.G. imaged eLACCO2.1 in vivo. C.T.V., F.B., A.B.L., H.H, and M.N. performed in vivo R-iLACCO imaging. Y.N. and R.E.C. supervised research and wrote the manuscript.

## Competing interests

The authors declare no competing interests.
