## [Peer Review File · Nature Communications]

REVIEWER COMMENTS

Reviewer #1 (Remarks to the Author):

- What are the noteworthy results?

The noteworthy results are the development of orthogonal fluorescent markers that can be used in vitro, brain slices and in vivo to label intracellular and extracellular lactate. The combination of both intra- and extracellular will be invaluable to determine the exact role of lactate in brain metabolism. The controversy about the role of lactate and whether it is an integral part of brain metabolism (ANLS – Astrocyte to neuron lactate shuttle) or a by-product of brain metabolism (NALS - Neuron to Astrocyte lactate shuttle) is slowly resolving towards the ANLS BUT there are still unknowns because it was not possible to see lactate movements in vivo in the brain. Most arguments for or against each hypothesis were derived from modeling, cell cultures and brain slices. The problem with the last two techniques is that they require bath medium with an extremely high glucose concentration which is beyond what would be observed in the brain and leads to erroneous conclusions. NMR spectroscopy has also been used but the technique is not without caveats. In vivo microdialysis and electrochemical electrodes can provide in vivo extracellular measures of glucose and lactate but they only provide extracellular measures and the insertion of the probes disrupts the normal functioning of cells in the area of insertion. The addition of intra- and extra-cellular lactate biosensors will likely provide crucial evidence for the understanding of lactate role in brain metabolism. The experiments presented appear to show strong specificity and relatively intense fluorescence, both essential for in vivo studies.

The three R-iLACCO1 variants represent a complementary set that address the wide variety of intracellular lactate levels – they will be extremely useful to measure intracellular lactate levels in the brain under various conditions. These represent a major advance in the area of brain lactate measurements. The in vivo demonstration of the effect of peripheral insulin on hippocampal neurons using the extracellular lactate biosensor and the impact of whiskers stimulation on somatosensory neurons using the intracellular lactate biosensors was very convincing.

If there is one suggestion I could make for additional work, it would be a test of in vivo changes in intra- and extracellular lactate changes following either physiological stimulation (with the whiskers somatosensory preparation) or hippocampal electrical stimulation (or other stimulation paradigm) to show the combined measurements of intra- and extracellular lactate in the same brain area in vivo. The experiment using glioblastoma cells clearly indicated this possibility. On the other hand, the authors have demonstrated many crucial aspects of the duplex use of their lactate biosensors. I am not sure if additional studies should be part of the present submission unless the other reviewers or the editors feel that the manuscript would be significantly enhanced by this additional work.

Will the work be of significance to the field and related fields? There are other lactate biosensors but the originality of this new set of biosensors is due to the highly increased fluorescence (a very important

aspect for any in vivo experiments) and the selectivity and compatibility of the intra- and extracellular lactate biosensors. I foresee they will be highly used in the near future to test various aspects of the role of lactate in the brain.

- Does the work support the conclusions and claims, or is additional evidence needed? Yes it does.

- Are there any flaws in the data analysis, interpretation and conclusions? - Do these prohibit publication or require revision? None

- Is the methodology sound? Does the work meet the expected standards in your field? Yes

- Is there enough detail provided in the methods for the work to be reproduced? Yes

Figure 5 e) in-graph legends: insulin typos

Line 105 "eLACCO1 (ref. 18), » "ref" typo

Reviewer #2 (Remarks to the Author):

Nasu & Campbell et al developed a next generation extracellular green fluorescent lactate sensor and a new red fluorescent intracellular lactate sensor. Compared to the first generation, the new green eLACCO2.1 is darker in the unbound state and brighter in the lactate-bound state, resulting in a >3-fold improvement in dynamic range compared to eLACCO1.1. The leader and anchor were optimized for surface localization of eLACCO2.1, and the authors demonstrate it can detect extracellular lactate changes with culture cells, brain slices, and in vivo with mice. Three affinity variants of a red intracellular R-iLACCO sensor were also developed, which are the first red fluorescent lactate sensors. The authors compared R-iLACCOs with the other intracellular lactate sensors in solution and in cultured cells, showing the R-iLACCOs have similar or better dynamic range than the best green-yellow lactate sensor FiLa and generally improved performance. They also report in vivo detection of stimulation-dependent lactate production as a proxy for increased glycolysis in mouse brain and show it can be used in drosophila. Finally, they demonstrate two-color imaging with there new sensors, providing the ability to

observe fluxes that reveal potential transport or coupling of lactate levels across cytosol, mitochondria, and ER.

Overall, the manuscript was written well. The engineering rationale was well explained, and the characterization was very thorough including the two-photon action spectra for example. The side-by-side comparisons with the other lactate sensors is very useful, and the detection of endogenous responses to insulin and whisker stimulation were exciting to see. The work was really quite comprehensive and excellent, and there is only one concern and a few minor questions:

1) Even though the apparent Ca^{2+} affinity is lower than the expected extracellular Ca^{2+} concentration, it's still pretty high. If the extracellular Ca^{2+} were in the 1-2mM range, eLACCO2.1 would be 80-90% saturated. But the Ca^{2+} dependence is relatively steep, so it is a concern if extracellular Ca^{2+} levels fluctuate such as with high activity when some reports indicate a significant depletion. In practice how much does the eLACCO signal fluctuate within this extracellular calcium range in the apo and bound states?

Minor:

2) The optimization of the leader and anchor of eLACCO is very useful. Similar to iGluSnFR3, the C-terminal anchor was optimized first and subsequently the N-terminal leader was optimized for the best anchor. Is there any indication that a combinatorial library would be beneficial? Or in practice is it sufficient to treat them as relatively independent. It could be useful to comment on this given that optimization was so important.

3) It wasn't quite clear why photostability of eLACCO in 3g and R-iLACCO in 4f were quantified differently with time constant versus integrated fluorescence.

Reviewer #3 (Remarks to the Author):

Building on previous work, the authors describe the engineering, characterization, and deployment of a pair of lactate sensors. Over the past year, several single-color fluorescent biosensors for lactate have been reported. The authors have engineered a new version of an extracellular green lactate sensor with a larger fluorescence change as well as a red intracellular lactate sensor, which is a larger advance.

Comparisons to other intensity-based lactate sensors suggest that although the red R-iLaccos in this manuscript are improved by the narrowly defined criterion of dF/F (using zero lactate as the reference point), and they do have bigger responses in cultured cells, they are not practically improved for use in vivo. It is surprising that only 1-2% changes in dF/F are seen for the in vivo manipulations of Figure 5, particularly compared to the published Laconic dR/R responses for similar manipulations (pmid 32694692; ~3% for isoflurane puffs and ~1% for air puffs). Given the much bigger raw dF/F for R-iLacco (compared to the % changes in ratio for Laconic), these small responses probably result from mismatch between the concentration response of R-iLacco 1.1 and the actual in vivo concentration changes. It is unclear whether the sensor is nearly saturated or nearly fully desaturated. This may have something to do with the large differences (for unknown reasons) between the in-cell calibration and the in-vitro dose responses (also noted below), and also because the resting lactate is rarely close to the reference concentration of zero.

Other comments:

1. The authors claim that the R-iLaccos display a 'limited amount of photoactivation upon blue-light illumination' (line 304), but the effect of photoactivation on fluorescence (9% dF/F , Figure 4e) is much larger than that seen in the in vivo experiments (1-2% dF/F , Figure 5h), and is also not negligible compared to the in vitro neuron and astrocyte changes (31-49% in Extended Data Table 3) that are produced by strong manipulations. It should be made clear that caution is required in using the red sensors in combination with green sensors or optogenetic manipulations.

2. Intensiometric sensors with large dF/F can be good for detecting changes in lactate from time to time, but they are not particularly good for determining the actual concentration of lactate. Some caution is therefore required in limiting the interpretation of Figure 6 to lactate dynamics and not levels (e.g. lines 312-324); it should be made clear that, because of the different properties of the sensors used and the unknown baseline levels in the different compartments, neither the relative lactate concentrations nor even the relative size of the lactate transients is known from the signals. To reinforce this, it might be better just to make the red and green transients about the same size and to provide separate axes indicating the scale of the dF/F changes. In some ways, it might be more valuable to express the same sensor in the different sites and to perform parallel experiments, since these would give more comparable results.

3. The claim that the red sensor variants detect lactate dynamics over a 'broad range of concentrations' is too strong given the uncertainty over the sensors' midpoints; the small differences between them; and the fact that lactate concentrations are rarely close to zero (lines 305-307).

4. The authors should credit at least some of the substantial prior work on subcellular lactate metabolism (see, for instance, pmid 19805739), of which there is no mention in the manuscript.

5. As mentioned above, there is a large discrepancy between the in-cell calibration K_{app} for R-iLacco variants (680 μ M – 4 mM, Fig. 4C) and the in vitro titration with the same variants as purified proteins, which are puzzlingly ~ 1 order of magnitude higher affinity (74 μ M – 350 μ M, Extended Data Fig. 6B). This difference could be due to temperature-dependence (which should be characterized); it could also be due to sensor concentration-dependent dimerization of the sensor (since LldR is known to dimerize, PMID 18988622). It could also be due to failure to control cellular lactate during the in-situ calibration (in which case the values may be misleading); an inhibitor of lactate dehydrogenase might be a valuable addition to the inhibitor cocktail.

Response to reviewer comments:

We greatly appreciate the insightful and constructive comments. These comments have proven very helpful for improving the manuscript. To address the comments, we have revised our manuscript in a point-to-point manner as described on the following pages. We have also revised the numbering of supplementary figures and tables to meet the formatting requirements of *Nature Communications*, as summarized in the table below. Newly added figures are shown in blue.

New	Old	Title
Supplementary figures		
Fig. S1	Extended Data Fig. 1	eLACCO0.9 exhibits a faster response than eLACCO1.
Fig. S2	Extended Data Fig. 2	Development of affinity-tuned eLACCO2.1.
Fig. S3	Fig. S1	Sequence alignment of TTHA0766, cpGFP, eLACCO1.1, eLACCO2.1, and deLACCO1.
Fig. S4	Extended Data Fig. 3	In vitro characterization of deLACCO1 and R-diLACCO1.
Fig. S5	Fig. S2	Lineage of eLACCO and R-iLACCO variants.
Fig. S6	Extended Data Fig. 4	In vitro characterization of eLACCO2.1.
Fig. S7	Extended Data Fig. 5	Construction of the R-iLACCO prototype.
Fig. S8	Fig. S3	Sequence alignment of LldR, cpmApple, R-iLACCO1, R-iLACCO1.1, R-iLACCO1.2, and R-diLACCO1.
Fig. S9	Fig. S4	Development of R-iLACCO variants with lower L-lactate affinity.
Fig. S10	Extended Data Fig. 6	In vitro characterization of R-iLACCO1.
Fig. S11	Extended Data Fig. 7	In vitro characterization of R-iLACCO1.1 and R-iLACCO1.2.
Fig. S12	Fig. S5	Fluorescence imaging of the other currently-available intracellular L-lactate biosensors.
Fig. S13	Fig. S6	Inhibition of L-lactate transporter perturbs intracellular L-lactate and pH dynamics.
Fig. S14	Fig. S7	Imaging of R-iLACCO variants in various tissues.
Fig. S15		Fluorescence imaging of eLACCO2.1 in extracellular Ca ²⁺ concentration range from 1 to 2 mM.
Fig. S16		Temperature-dependent lactate response of R-iLACCO variants.
Tables		
Table 1	Extended Table 1	One- and two-photon photophysical parameters of eLACCO2.1.
Table 2	Extended Table 2	One- and two-photon photophysical parameters of R-iLACCO1.
Table 3	Extended Table 3	Comparison of performance for intracellular L-lactate biosensors.

Reviewer #1

- What are the noteworthy results?

The noteworthy results are the development of orthogonal fluorescent markers that can be used in vitro, brain slices and in vivo to label intracellular and extracellular lactate. The combination of both intra- and extracellular will be invaluable to determine the exact role of lactate in brain metabolism. The controversy about the role of lactate and whether it is an integral part of brain metabolism (ANLS – Astrocyte to neuron lactate shuttle) or a by-product of brain metabolism (NALS - Neuron to Astrocyte lactate shuttle) is slowly resolving towards the ANLS BUT there are still unknowns because it was not possible to see lactate movements in vivo in the brain. Most arguments for or against each hypothesis were derived from modeling, cell cultures and brain slices. The problem with the last two techniques is that they require bath medium with an extremely high glucose concentration which is beyond what would be observed in the brain and leads to erroneous conclusions. NMR spectroscopy has also been used but the technique is not without caveats. In vivo microdialysis and electrochemical electrodes can provide in vivo extracellular measures of glucose and lactate but they only provide extracellular measures and the insertion of the probes disrupts the normal functioning of cells in the area of insertion. The addition of intra- and extra-cellular lactate biosensors will likely provide crucial evidence for the understanding of lactate role in brain metabolism. The experiments presented appear to show strong specificity and relatively intense fluorescence, both essential for in vivo studies.

The three R-iLACCO1 variants represent a complementary set that address the wide variety of intracellular lactate levels – they will be extremely useful to measure intracellular lactate levels in the brain under various conditions. These represent a major advance in the area of brain lactate measurements. The in vivo demonstration of the effect of peripheral insulin on hippocampal neurons using the extracellular lactate biosensor and the impact of whiskers stimulation on somatosensory neurons using the intracellular lactate biosensors was very convincing.

Thank you very much for your comprehensive overview of the background of this study. We also appreciate your positive comments on this work.

If there is one suggestion I could make for additional work, it would be a test of in vivo changes in intra- and extracellular lactate changes following either physiological stimulation (with the whiskers somatosensory preparation) or hippocampal electrical stimulation (or other stimulation paradigm) to show the combined measurements of intra- and extracellular lactate in the same brain area in vivo. The experiment using glioblastoma cells clearly indicated this possibility. On the other hand, the authors have demonstrated many crucial aspects of the duplex use of their lactate biosensors. I am not sure if additional studies should be part of the present submission unless the other reviewers or the editors feel that the manuscript would be significantly enhanced by this additional work.

Thank you very much for valuable suggestion. *In vivo* imaging of intra- and extracellular lactate changes simultaneously is one of the most important and interesting future applications for the LACCO biosensors, but is beyond the scope of the current study. Given the general enthusiasm for our work in its current form, we have elected to leave these time-consuming additional studies for a future manuscript.

Will the work be of significance to the field and related fields? There are other lactate biosensors but the originality of this new set of biosensors is due to the highly increased fluorescence (a very important aspect for any in vivo experiments) and the selectivity and compatibility of the intra- and extracellular

lactate biosensors. I foresee they will be highly used in the near future to test various aspects of the role of lactate in the brain.

Thank you for your affirmative comments for this work.

- Does the work support the conclusions and claims, or is additional evidence needed? Yes it does.
- Are there any flaws in the data analysis, interpretation and conclusions? - Do these prohibit publication or require revision? None
- Is the methodology sound? Does the work meet the expected standards in your field? Yes
- Is there enough detail provided in the methods for the work to be reproduced? Yes

Thank you for your supportive evaluations on this work.

Figure 5 e) in-graph legends: insulin typos
Line 105 “eLACCO1 (ref. 18), » “ref” typo

Thank you for pointing these out. We have fixed the insulin typo. We believe that (ref. 18) is the correct way to write this particular citation in *Nature Communications* format, due to the fact that it is following a number. We will of course follow the suggestions of the editor on this point.

Reviewer #2

Nasu & Campbell et al developed a next generation extracellular green fluorescent lactate sensor and a new red fluorescent intracellular lactate sensor. Compared to the first generation, the new green eLACCO2.1 is darker in the unbound state and brighter in the lactate-bound state, resulting in a >3-fold improvement in dynamic range compared to eLACCO1.1. The leader and anchor were optimized for surface localization of eLACCO2.1, and the authors demonstrate it can detect extracellular lactate changes with culture cells, brain slices, and in vivo with mice. Three affinity variants of a red intracellular R-iLACCO sensor were also developed, which are the first red fluorescent lactate sensors. The authors compared R-iLACCOs with the other intracellular lactate sensors in solution and in cultured cells, showing the R-iLACCOs have similar or better dynamic range than the best green-yellow lactate sensor FiLa and generally improved performance. They also report in vivo detection of stimulation-dependent lactate production as a proxy for increased glycolysis in mouse brain and show it can be used in drosophila. Finally, they demonstrate two-color imaging with there new sensors, providing the ability to observe fluxes that reveal potential transport or coupling of lactate levels across cytosol, mitochondria, and ER.

Overall, the manuscript was written well. The engineering rationale was well explained, and the characterization was very thorough including the two-photon action spectra for example. The side-by-side comparisons with the other lactate sensors is very useful, and the detection of endogenous responses to insulin and whisker stimulation were exciting to see. The work was really quite comprehensive and excellent, and there is only one concern and a few minor questions:

Thank you very much for your positive comments for this work.

1) Even though the apparent Ca^{2+} affinity is lower than the expected extracellular Ca^{2+} concentration, it's still pretty high. If the extracellular Ca^{2+} were in the 1-2 mM range, eLACCO2.1 would be 80-90% saturated. But the Ca^{2+} dependence is relatively steep, so it is a concern if extracellular Ca^{2+} levels fluctuate such as with high activity when some reports indicate a significant depletion. In practice how much does the eLACCO signal fluctuate within this extracellular calcium range in the apo and bound states?

We appreciate your comments on the Ca^{2+} dependency of eLACCO2.1. To address this concern, we recorded the fluorescence intensity of eLACCO2.1-expressing HeLa cells in the Ca^{2+} concentration range from 1 to 2 mM, in the presence and absence of 10 mM L-lactate. We have now added the *in situ* Ca^{2+} titration data in Supplementary Figure 15 and revised the main text (lines 362–370) as, “An important consideration for the application of eLACCO2.1 is its Ca^{2+} dependent fluorescence response. For extracellular applications this should not be a major concern because the apparent K_d of eLACCO2.1 for Ca^{2+} is substantially lower than the physiological or pathological Ca^{2+} concentration range (~1.5-1.7 mM) in the extracellular space of brain tissue (Fig. 3e)²¹. However, the Ca^{2+} concentration in the extracellular environment can transiently decrease to ~1 mM during neural activity evoked by extreme stimulations such as long trains of experimentally induced action potentials³⁴. We determined that a change in extracellular Ca^{2+} concentration from 2 mM to 1 mM could cause a -9% and -5% fluorescence intensity for lactate-bound and lactate-unbound eLACCO2.1, respectively (Supplementary Fig. 15).”

Minor:

2) The optimization of the leader and anchor of eLACCO is very useful. Similar to iGluSnFR3, the C-terminal anchor was optimized first and subsequently the N-terminal leader was optimized for the best anchor. Is there any indication that a combinatorial library would be beneficial? Or in practice is it sufficient to treat them as relatively independent. It could be useful to comment on this given that optimization was so important.

Thank you very much for comments on the optimization of the leader and anchor of eLACCO biosensor. We have not tried the combinatorial library to identify the best combination at a time, but our results suggest that the C-terminal anchor has a larger impact on the efficiency of cell surface localization than N-terminal leader, as described in Supplementary Note 3. To comment on the potential benefit of the combinatorial library screening, we have now added the following text to Supplementary Note 3, “In this study we separately screened the anchor domains first followed by the leader sequences second. It is possible that screening of a combinatorial library of leader and anchor might provide another optimal combination.”

3) It wasn't quite clear why photostability of eLACCO in 3g and R-iLACCO in 4f were quantified differently with time constant versus integrated fluorescence.

We appreciate your comments on the quantification of photostability. GFP, and the GFP-based biosensors developed in this work, show a simple exponential decay of fluorescence intensity upon strong excitation illumination (Figure 3g). This data enabled us to obtain time constants (τ) by curve fitting. On the other hand, mApple RFP and mApple-based biosensors show more complex fluorescence decays upon strong excitation illumination (Figure 4f), hindering a simple curve fitting to obtain τ . This complex photophysics of mApple has been reported in a previous report (Dean et al. *Biophys. J.* **101**, 961–969 (2011)). Accordingly, we quantified the photostability of R-iLACCO

biosensors by integrating fluorescence intensities as an alternative to τ . To explain the reason why we used the integrated fluorescence for R-iLACCO variants instead of τ_{bleach} used for eLACCO2.1, we have now added the following text to the main text (lines 264–266), “*To assess the photostability of R-iLACCO variants, we used the integrated fluorescence (IF) in HeLa cells (rather than τ_{bleach}), to better account for complex photobleaching decays of mApple-based biosensors²⁸.*”

Reviewer #3

Building on previous work, the authors describe the engineering, characterization, and deployment of a pair of lactate sensors. Over the past year, several single-color fluorescent biosensors for lactate have been reported. The authors have engineered a new version of an extracellular green lactate sensor with a larger fluorescence change as well as a red intracellular lactate sensor, which is a larger advance.

Thank you for your comments to summarize this work in the fields of lactate biosensor engineering.

Comparisons to other intensity-based lactate sensors suggest that although the red R-iLaccos in this manuscript are improved by the narrowly defined criterion of dF/F (using zero lactate as the reference point), and they do have bigger responses in cultured cells, they are not practically improved for use in vivo. It is surprising that only 1-2% changes in dF/F are seen for the in vivo manipulations of Figure 5, particularly compared to the published Laconic dR/R responses for similar manipulations (pmid 32694692; $\sim 3\%$ for isoflurane puffs and $\sim 1\%$ for air puffs). Given the much bigger raw dF/F for R-iLacco (compared to the % changes in ratio for Laconic), these small responses probably result from mismatch between the concentration response of R-iLacco 1.1 and the actual in vivo concentration changes. It is unclear whether the sensor is nearly saturated or nearly fully desaturated. This may have something to do with the large differences (for unknown reasons) between the in-cell calibration and the in-vitro dose responses (also noted below), and also because the resting lactate is rarely close to the reference concentration of zero.

Thank you very much for your comments. To more clearly discuss the potential reason why R-iLACCO1.1 showed a relatively limited response *in vivo*, we have now added the following text to the main text (lines 397–402), “*The fluorescence response ($\Delta F/F \sim 1\%$) of R-iLACCO1.1 in the somatosensory cortex of whisker stimulated mice is comparable to that of Laconic ($\Delta R/R \sim 1\%$) under similar conditions³⁸. Based on the fact that the $\Delta F/F$ of R-iLACCO1.1 in cultured cells is much larger than $\Delta R/R$ of Laconic, a substantially larger response could have been expected. The reason for this discrepancy is unclear, but one possible explanation is that the lactate affinity of R-iLACCO1.1 might not be optimal for detecting the changes that occur under these particular stimulation conditions.*”

Other comments:

1. The authors claim that the R-iLaccos display a ‘limited amount of photoactivation upon blue-light illumination’ (line 304), but the effect of photoactivation on fluorescence (9% dF/F , Figure 4e) is much larger than that seen in the in vivo experiments (1-2% dF/F , Figure 5h), and is also not negligible compared to the in vitro neuron and astrocyte changes (31-49% in Extended Data Table 3) that are produced by strong manipulations. It should be made clear that caution is required in using the red sensors in combination with green sensors or optogenetic manipulations.

Thank you very much for helpful comments on the blue-light photoactivation of R-iLACCO biosensors. To better describe a need for caution regarding the photoactivation of R-iLACCO biosensors in lactate imaging in cultured cells and *in vivo*, we have now revised the main text (lines 385–386) as, “We also confirmed that R-iLACCO1 and its affinity variants display no undesirable punctate intracellular accumulation (**Fig. 4g–j**).”, and added the following text to the main text (lines 407–453), “Multiplexed imaging of R-iLACCO biosensors with green fluorescent biosensors, or combined use with optogenetic tools, should be performed with caution since blue-light photoactivation ($\Delta F/F \sim 9\%$, **Fig. 4e**) of R-iLACCO biosensors is non-negligible in some applications of cultured cells (**Fig. 4h–j**) and *in vivo* (**Fig. 5f–i**). Parallel experiments with the control biosensor R-diLACCO1 would be recommended in the multiplexed applications.”

2. Intensiometric sensors with large dF/F can be good for detecting changes in lactate from time to time, but they are not particularly good for determining the actual concentration of lactate. Some caution is therefore required in limiting the interpretation of Figure 6 to lactate dynamics and not levels (e.g. lines 312-324); it should be made clear that, because of the different properties of the sensors used and the unknown baseline levels in the different compartments, neither the relative lactate concentrations nor even the relative size of the lactate transients is known from the signals. To reinforce this, it might be better just to make the red and green transients about the same size and to provide separate axes indicating the scale of the dF/F changes. In some ways, it might be more valuable to express the same sensor in the different sites and to perform parallel experiments, since these would give more comparable results.

Thank you very much for your valuable comments. We agree that our imaging data in Figure 6 does not provide the information on the relative lactate concentration and the relative size of lactate transients between cell compartments. To better describe the lactate dynamics, we have now revised $\Delta F/F$ graphs in Figure 6c,f,i by adding the second axes to separately represent $\Delta F/F$ in green and red channel. We have checked the text to confirm that we have appropriately interpreted these results.

3. The claim that the red sensor variants detect lactate dynamics over a ‘broad range of concentrations’ is too strong given the uncertainty over the sensors’ midpoints; the small differences between them; and the fact that lactate concentrations are rarely close to zero (lines 305-307).

Thank you for your comments. To better describe the utility of the lactate affinity variants for R-iLACCO, we have now revised the main text (lines 386–388) as, “In addition, the availability of multiple affinity variants may be helpful for investigating intracellular L-lactate dynamics in different cell types with various basal lactate levels³⁷.”

4. The authors should credit at least some of the substantial prior work on subcellular lactate metabolism (see, for instance, pmid 19805739), of which there is no mention in the manuscript.

We appreciate your suggestion. To better credit the previous works on subcellular lactate metabolism, we have now added the following text to the main text (lines 458–459), “This is consistent with previous reports that suggest that lactate can shuttle between cytosol and mitochondria³⁹, and cytosol and ER¹¹.”

5. As mentioned above, there is a large discrepancy between the in-cell calibration K_{app} for R-iLacco variants (680 μM – 4 mM, Fig. 4C) and the in vitro titration with the same variants as purified proteins, which are puzzlingly ~ 1 order of magnitude higher affinity (74 μM – 350 μM , Extended Data Fig. 6B). This difference could be due to temperature-dependence (which should be characterized); it could also be due to sensor concentration-dependent dimerization of the sensor (since LldR is known to dimerize, pmid 18988622). It could also be due to failure to control cellular lactate during the in-situ calibration (in which case the values may be misleading); an inhibitor of lactate dehydrogenase might be a valuable addition to the inhibitor cocktail.

To investigate the possibility that temperature dependence is the reason for the discrepancy of K_{app} of R-iLACCO variants between purified proteins and in cells, we now include data for lactate titrations at 25 and 37 $^{\circ}\text{C}$ (Supplementary Figure 16), and we have added the following sentences to the main text (lines 388–394), “*The apparent K_{ds} of R-iLACCO variants in cells (Fig. 4c) are much higher than those measured with purified proteins (Supplementary Fig. 10b). Considering a change in temperature (37 $^{\circ}\text{C}$ used for cells vs. 25 $^{\circ}\text{C}$ for purified proteins) does not have a substantial impact on apparent K_{ds} of R-iLACCO variants (Supplementary Fig. 16), this discrepancy must be due to molecular crowding or other yet-unidentified environmental differences between the purified proteins in buffer and the protein expressed in mammalian cells.*”

REVIEWERS' COMMENTS

Reviewer #2 (Remarks to the Author):

All my comments were well addressed, and my enthusiasm for this work remains high.

Reviewer #3 (Remarks to the Author):

The authors have done well in this revision to address concerns related to (1) the midpoint of their red sensor, (2) additional variables (temperature and photoswitching) that complicate sensor interpretation, and (3) the sensors' context with some additional experiments and several textual changes.

First, the authors have provided additional characterization of their novel lactate sensors, particularly with respect to Ca^{2+} and temperature dependences. Second, they have better contextualized their new sensors in light of previous *in vivo* results using Laconic and of previous work studying compartmentalized lactate metabolism. Third, some of the limitations of the sensors have been discussed at greater length, which should guide users in deploying the sensors more carefully.

Dear reviewers,

We greatly appreciate the approval from the reviewers. Based on the editor's additional communication with the reviewers, and discussion within the editorial team, we have been requested to include a brief discussion of the intracellular lactate lifetime sensor, LiLac. To address this comment, we have now revised Introduction and added the specifications of LiLac to Table 3.

Reviewer #2

All my comments were well addressed, and my enthusiasm for this work remains high.

Thank you. We appreciate all of your comments on the revision of this manuscript.

Reviewer #3

The authors have done well in this revision to address concerns related to (1) the midpoint of their red sensor, (2) additional variables (temperature and photoswitching) that complicate sensor interpretation, and (3) the sensors' context with some additional experiments and several textual changes.

First, the authors have provided additional characterization of their novel lactate sensors, particularly with respect to Ca^{2+} and temperature dependences. Second, they have better contextualized their new sensors in light of previous *in vivo* results using Laconic and of previous work studying compartmentalized lactate metabolism. Third, some of the limitations of the sensors have been discussed at greater length, which should guide users in deploying the sensors more carefully.

Thank you. We are grateful for all of your suggestions on the revision of this manuscript.